# Dynamic control of IDP interaction network via diverse binding pathways

**Jae-Yeol Kim** ⓘ **& Hoi Sung Chung** ⓘ ✉

Binding promiscuity is a central feature of interactions involving intrinsically disordered proteins (IDPs). IDPs can interact even simultaneously with multiple binding partners, but quantitative characterization of these multicomponent interactions is challenging. Here, we characterize the binding pathways of the transactivation domain (TAD) of p53 with two binding partners (Taz2 and Mdm2) using three-color single-molecule Förster resonance energy transfer (FRET) spectroscopy. We show that the interactions of these three proteins occur via two pathways. The first pathway is competitive in that binding of one partner occurs after the other partner completely dissociates. The second is an allosteric pathway via the formation of a ternary complex. High time-resolution FRET using photon-by-photon analysis shows that these heterogeneous three-component interaction pathways are closely related with diverse transition paths of two-component TAD-Taz2 binding. Kinetic analysis shows that the allosteric pathway allows faster exchange of the binding partners with opposite functions. Our work demonstrates how a heterogeneous allosteric binding network can enable a faster response to changes in the external environment.

Intrinsic disorder is prevalent in the proteome and is involved in many biological processes[1], such as coupled folding and binding[2], signaling[3], formation of biomolecular condensates[4,5], and protein aggregation[6]. The conformational flexibility of intrinsically disordered proteins (IDPs) allows them to interact with multiple partners, making them hub proteins in complex binding networks[7]. These interactions can occur simultaneously with more than one binding partner when an IDP has multiple binding sites. This type of interaction is a unique feature of IDPs that allows for cooperative and allosteric control of various functions[8,9].

The N-terminal disordered domain (transactivation domain, TAD) of the tumor suppressor protein p53 recruits numerous proteins that are required for tight control of its function as a transcription factor[10]. The murine double minute 2 (Mdm2), E3 ubiquitin ligase, is a negative regulator of p53, which is responsible for p53 degradation[11]. The TAD also binds the transcriptional adapter zinc-binding domain 2 (Taz2) of the CREB-binding protein (CBP), a transcriptional coactivator[12,13]. Under normal conditions, the TAD is bound to Mdm2, keeping the p53

level low[14]. On the other hand, under stressed situations, TAD is phosphorylated[15,16], and Mdm2 is replaced by other proteins to activate many stress response genes[10]. Phosphorylation of serine and threonine residues in TAD reduces the Mdm2 affinity[13,17–19] and increases the affinity of the p300/CBP coactivator domains, including Taz2[13,19,20].

TAD has two subdomains, AD1 and AD2 (Fig. 1a), which interact with different binding partners (see Supplementary Fig. 1 for the amino acid sequences of the proteins). Mdm2 binds to AD1[11], whereas Taz2 binds both AD1 and AD2, with a 500 times higher affinity for AD2[13,21]. Ferreon et al.[13] have shown that these three proteins can form a ternary complex, where AD1 binds Mdm2 and AD2 binds Taz2. This result suggests that the binding and exchange of the two binding partners of TAD occur in a way that is more complex than simple competitive binding, which may also be implicated in the mechanism of p53 function and control. TAD is a useful model system to study complex interactions involving two binding sites[22].

In this work, we employ three-color single-molecule Förster resonance energy transfer (FRET) spectroscopy to understand the

Laboratory of Chemical Physics, National Institute of Diabetes and Digestive and Kidney Diseases, National Institutes of Health, Bethesda, MD, USA.
✉e-mail: chunghoi@niddk.nih.gov

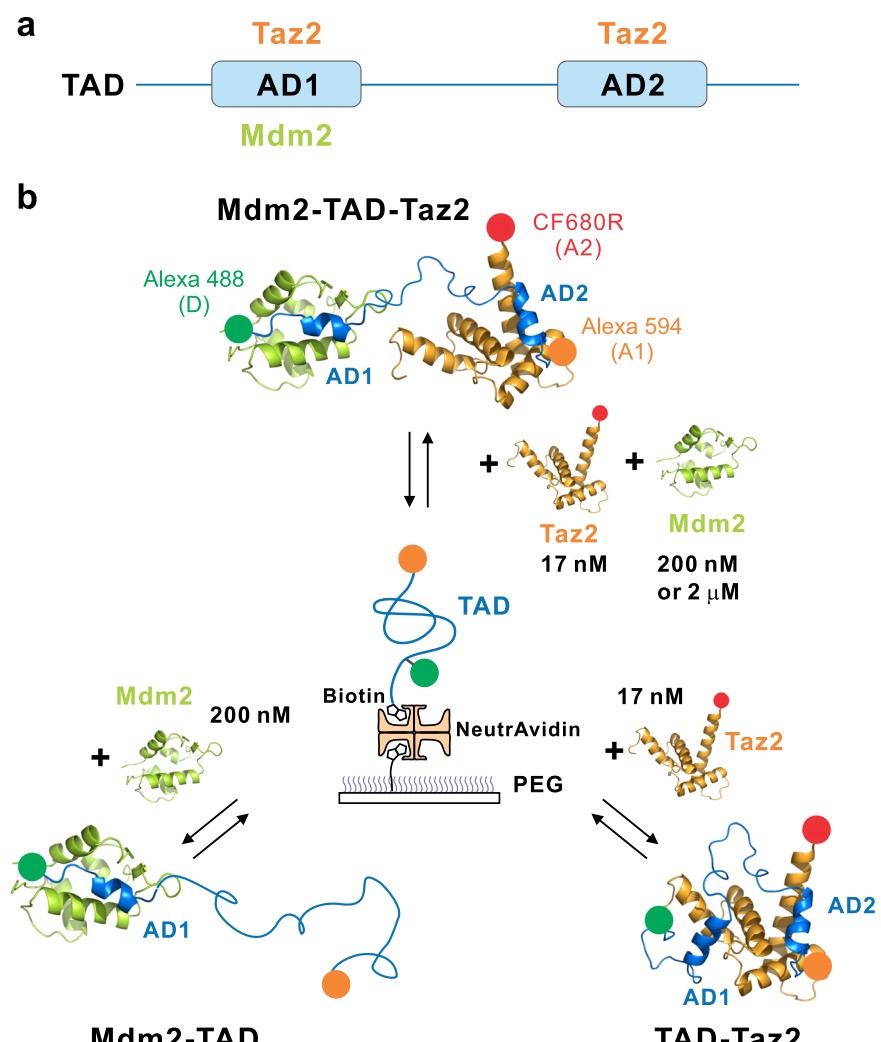

**Fig. 1 | Binary and ternary complex formation of TAD with Mdm2 and Taz2.**
**a** Two binding sites, AD1 and AD2, in TAD and their interactions with two binding partners, Taz2 and Mdm2. Taz2 interacts with both sites, while Mdm2 interacts with only AD1. **b** Schematic illustration of single-molecule FRET experiments. TAD is labeled with donor (D, Alexa 488) and acceptor 1 (A1, Alexa 594) and immobilized on a polyethylene glycol (PEG)-coated glass surface (middle). TAD is then incubated with (1) acceptor 2 (A2, CF680R)-labeled Taz2 (17 nM) (lower right), (2) unlabeled Mdm2 (200 nM) (lower left), or (3) unlabeled Mdm2 (200 nM or 2 μM) and A2-labeled Taz2 (17 nM) (upper) to study the formation of TAD-Taz2, TAD-Mdm2, and ternary complexes, respectively.

complete mechanism and structural origin of three-component interactions. We attached the donor (D, Alexa 488) and acceptor 1 (A1, Alexa 594) to the N- and C-termini of TAD, respectively, and attached acceptor 2 (A2, CF680R) to the N-terminus of Taz2. This allows us to monitor the conformational change of TAD upon binding of Taz2. Although Mdm2 is not labeled, its binding can be indirectly monitored by changes in the relative fluorescence intensity of the three dyes. We immobilized TAD on a polyethylene glycol (PEG)-coated glass surface (Fig. 1b) and performed three types of experiments: (1) TAD-Taz2 binding by incubating with Taz2 (17 nM, 15 nM of A2-labeled Taz2, 88% labeling efficiency), (2) TAD-Mdm2 binding by incubating with Mdm2 (200 nM), and (3) binding of the three components by incubating with both Taz2 (17 nM) and Mdm2 (200 nM or 2 μM) (Fig. 1b). The effect of immobilization and dye labeling seems minimal as the measured dissociation constants are similar to previously measured values[12,13,17–19] (Supplementary Table 1).

## Results

### Two pathways of binding partner exchange

In our experiments, the donor was excited by a continuous-wave laser (485 nm). After excitation, the energy is transferred from D to A1 and subsequently to A2 or directly from D to A2, depending on the

distance between the three dye pairs. Different conformational states are represented by the fractions of acceptor 1 ($\varepsilon_1 = n_{A1}/n$) and acceptor 2 ($\varepsilon_2 = n_{A2}/n$). Here, $n_{A1}$, $n_{A2}$, and $n_D$ are the photon count rates in D, A1, and A2 channels, respectively, and $n$ (= $n_{A1} + n_{A1} + n_D$) is the total count rate.

Figure 2a shows typical binned trajectories (20 ms bin time) of the two-component TAD-Taz2 binding experiment, which was performed as a reference for the three-component experiment. More trajectories are displayed in Supplementary Fig. 2a. The states can be assigned according to the relative intensity of the three fluorophore detection channels, which are indicated by the color bars above the trajectories. There are three states. The state with the highest A2 intensity (red bars) is the bound state of TAD and Taz2. This bound state is compact because both AD1 and AD2 of TAD interact with Taz2[21], and most of the energy is transferred to A2. In the unbound state (blue bar), the TAD is extended due to electrostatic repulsion[23] and the FRET efficiency from the donor to A1 is lower, resulting in the similar count rates in the D and A1 channels. Note that there is no A2 in the unbound TAD. The non-zero A2 signal results from A1 photons leaking into the A2 channel (i.e., cross-talk, ~20%). In addition to the 3-color bound state and the unbound state, there is a third state, in which the A1 count rate is highest (yellow bar). This state is a bound state without active A2,

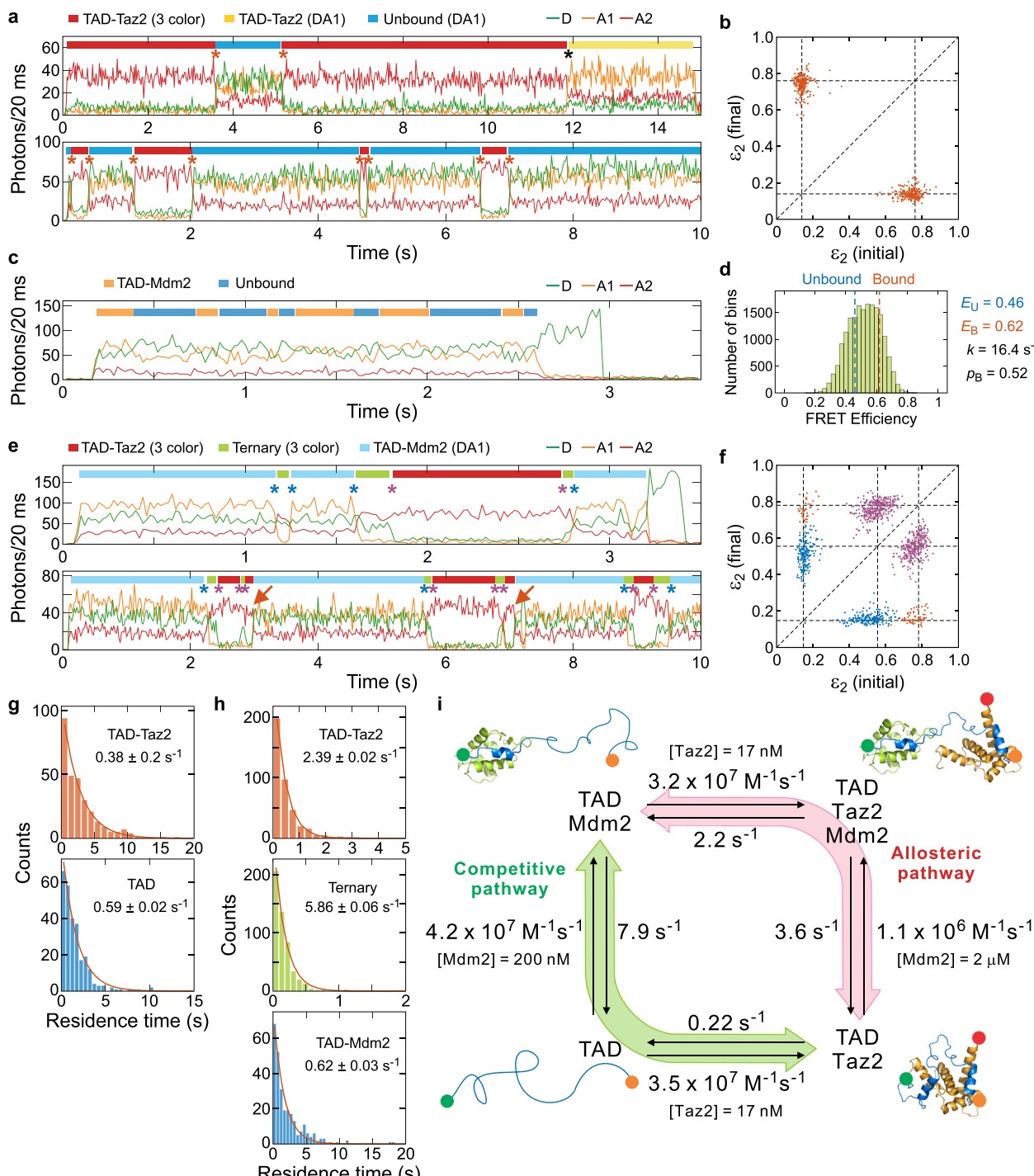

**Fig. 2 | Three-color FRET of binary and ternary complex formation of TAD with Mdm2 and Taz2. a** Representative binned (20 ms bin time) fluorescence trajectories of three-color FRET experiments of TAD-Taz2 binding. [Taz2] = 17 nM. The segments of different protein states and combinations of active dyes are indicated by different color bars above the trajectories. Transitions between the bound and unbound states are indicated by red asterisks. A black asterisk indicates photobleaching of A2 in the bound state. **b** Two-dimensional transition map constructed with acceptor 2 fractions ($\varepsilon_2$) before and after binding and dissociation transitions of TAD and A2-active Taz2. **c** A representative binned fluorescence trajectory (20 ms bin time) of the two-color FRET experiment (detected in three channels) of TAD-Mdm2 binding. [Mdm2] = 200 nM. **d** A histogram of the apparent FRET efficiency calculated for each bin. The FRET efficiencies of the bound and unbound states ($E_B$ and $E_U$), the relaxation rate ($k$), and the fraction of the bound state ($p_B$) were determined using a two-state maximum likelihood analysis of photon trajectories without binning (see "Methods"). **e** Representative binned (20 ms bin

time) fluorescence trajectories of three-color FRET experiments for the ternary complex formation. [Taz2] = 17 nM and [Mdm2] = 2 μM. Blue and purple asterisks indicate transitions between the TAD-Mdm2 complex and the ternary complex and between the ternary complex and the TAD-Taz2 complex, respectively. Direct transitions between the TAD-Taz2 and TAD-Mdm2 complexes are indicated by red arrows. **f** Transitions in (**e**) are shown in the two-dimensional transition map of $\varepsilon_2$ with corresponding colors. **g** Waiting time distributions in the TAD-Taz2 bound state and unbound state. Red lines indicate exponential fitting. **h** The waiting time distributions in the three states in the ternary complex formation experiments. **i** Double-pathway model and rate constants. 201, 58, and 130 trajectories were analyzed for the TAD-Taz2 binding, TAD-Mdm2 binding, and ternary complex formation experiments, respectively. All measurements were performed in pH 7, 20 mM Tris-HCl with 15 mM NaCl. See Supplementary Fig. 2 for additional example trajectories. Source data for (**b**) and (**f**) are provided as a Source Data file.

which appears due to either incomplete A2 labeling or photobleaching of A2. The state that immediately follows the 3-color bound state in the upper trajectory in Fig. 2a (transition indicated by a black asterisk) is the latter case. Again, the signal in the A2 channel results from the leakage of A1 photons. The transitions between the unbound state and the bound state with active A2 are indicated with red asterisks in the trajectories in Fig. 2a. Alternation between the bound (3-color or 2-color) state and the unbound state indicates the reversible binding and dissociation of Taz2. A transition map constructed using the $\varepsilon_2$ values before and after binding and dissociation of A2-active Taz2 in Fig. 2b shows a clear two-state behavior. From the waiting time distributions in the bound and unbound states, the dissociation and binding rates are determined, respectively (Fig. 2g).

Figure 2c shows a trajectory of the TAD-Mdm2 binding experiment (see Supplementary Fig. 2b for more trajectories). This is a two-color experiment because Mdm2 is unlabeled. It is noticeable that there are transitions between a state with a higher A1 signal and another state with a higher D signal, corresponding to the Mdm2-bound (orange bar) and unbound (blue bar) states, respectively. However, it is not possible to accurately determine the waiting times due to the small FRET efficiency difference (two distributions in the FRET efficiency histogram significantly overlap in Fig. 2d). Therefore, we used a two-state maximum likelihood analysis of photon trajectories[24] (see *Determination of binding kinetics of TAD-Mdm2 using the maximum likelihood method* in "Methods") to extract the binding and dissociation rates listed in Fig. 2d.

Figure 2e shows the representative trajectories of the experiment of all three proteins mixed together (see Supplementary Fig. 2c for more trajectories). In this experiment, the Mdm2 concentration (2 μM) is 10 times higher than the dissociation constant ($K_d = 186$ nM, Supplementary Table 1). Under this condition, free, unbound TAD is not detected in binned trajectories because of its low population and short dwell time, which simplified the interpretation and analysis of the data. We also performed the same experiment at a lower Mdm2 concentration of 200 nM, which resulted in similar kinetic parameters (See Supplementary Fig. 3 and Table 2 for fluorescence trajectories and analysis results with [Mdm2] = 200 nM). In Fig. 2e, TAD-Mdm2 bound state is indicated by a light blue bar. The A1 signal is slightly higher than the D signal, consistent with the observation from the TAD-Mdm2 binding experiment shown in Fig. 2c. The state with the highest A2 signal (red bar) is the TAD-Taz2 bound state as observed in the TAD-Taz2 experiment (Fig. 2a). Notably, most transitions between the TAD-Taz2 bound state and the TAD-Mdm2 bound state occur via an intermediate state (light green bar). This state exhibits similar levels of the D and A2 signals and almost no A1 signal. The extremely low A1 signal indicates that A1 and A2 are close, similar to the TAD-Taz2 bound state, suggesting that the C-terminus of TAD (i.e., AD2), where A1 is attached, is bound to Taz2. On the other hand, the D signal is much higher than that of the TAD-Taz2 bound state, indicating that D is far from A1 and A2. This is made possible by the binding of Mdm2 to the N-terminal AD1 subdomain of TAD. Therefore, this new state corresponds to a ternary complex of Mdm2, TAD, and Taz2, which mediates the exchange of the two binding partners, Taz2 and Mdm2 (see the structures in Figs. 1b and 2i). Since Taz2 has a much higher affinity for AD2 than for AD1[13], AD1 can detach from Taz2 transiently while AD2 remains bound, so that Mdm2 can bind to AD1 to form the ternary complex.

The transitions involving the ternary complex are shown in the transition map in Fig. 2f that are constructed using the $\varepsilon_2$ values before and after transitions. Transitions between the ternary complex and the TAD-Mdm2 state (blue dots) and between the ternary complex and the TAD-Taz2 state (purple dots) are clearly observed. Combined, these transitions correspond to the exchange of Taz2 and Mdm2 via formation of the ternary complex. Interestingly, the transition map shows an additional cluster corresponding to direct transitions between the

TAD-Taz2 and TAD-Mdm2 states, which do not involve the ternary complex (red dots). These transitions are indicated by red arrows on the lower trajectory in Fig. 2e.

The example trajectories and transition maps show that there are two different pathways that connect the two binary complexes, TAD-Mdm2 and TAD-Taz2. Based on these observations, we constructed a kinetic model shown in Fig. 2i. One pathway for exchanging the binding partners involves the formation of the ternary complex (magenta pathway in Fig. 2i). We refer to this pathway as an allosteric pathway because two binding partners may affect each other's binding (see Discussion). In the second pathway, the exchange of the binding partners occurs with complete dissociation of one partner, followed by association of the other. This pathway does not involve the ternary complex, and there is no allosteric effect. We refer to this pathway as a competitive pathway (green pathway in Fig. 2i). This pathway involves free, disordered TAD, but it was not observed in binned trajectories due to the high Mdm2 concentration, as mentioned above. A maximum likelihood analysis of photon trajectories near the transitions determined that the FRET efficiency of a short segment (~12 ms) of the TAD-Mdm2 state right before the transition to TAD-Taz2 is 0.48, very close to that of free TAD in the TAD-Mdm2 binding experiment above (Fig. 2d). This result verifies that the competitive pathway involves free TAD. A similar result was obtained from the experiment with [Mdm2] = 200 nM (see *Calculation of the relative flux involving the ternary complex* in "Methods" and Supplementary Table 3).

Using the waiting time distributions in the states and the maximum likelihood analysis for TAD-Mdm2 binding, we determined all rate constants (see *Determination of rate constants* in "Methods") as shown in Fig. 2i and summarized in Supplementary Table 2.

## Heterogeneous binding transition paths

Next, we sought the structural origin of the two exchange pathways. We hypothesized that this heterogeneity is encoded in the nature of the complex transition paths (TPs), especially those of TAD-Taz2 binding, because they involve interactions of both AD1 and AD2. In contrast, Mdm2 binding simply involves the formation of a single α helix in AD1. The transition path is a tiny fraction of a molecular trajectory that connects two states (e.g., the bound and unbound states in the case of binding), which contains mechanistic structural information about the process[25–27]. However, the duration of the TP is generally very short compared to the waiting time (i.e., inverse of the rate constant) in each state, and thus, it is more difficult to measure. We previously observed heterogeneous TPs of binding of TAD to another binding partner NCBD using three-color FRET[28]. TPs could be clustered into two groups with different heterogeneity and TP times, ranging from 10 μs to 1 ms. The order of interaction of different TAD subdomains with NCBD varied depending on the TP. If the binding transition paths of the formation of the TAD-Taz2 complex are similarly heterogeneous, this may be related to the two exchange pathways of Taz2 and Mdm2. To measure the short TP times, the experiment was performed at an illumination intensity 100 times higher (average photon count rates of ~300 ms$^{-1}$) than that used to collect the data in Fig. 2, and a photon-by-photon maximum likelihood analysis was used[28,29].

Figure 3a shows three example binding and dissociation transitions of TAD and Taz2 collected under this condition. Since the transient TPs are not readily visible in these binned trajectories (200 μs bin time), we analyzed the corresponding photon trajectory of each transition (Fig. 3c) using the maximum likelihood method with a one-intermediate model (i.e., three-state model)[28], consisting of a bound state, an unbound state, and an intermediate state that represents the TP (Fig. 3b) (see *Maximum likelihood method for the transition path analysis* in "Methods"). From the maximum of the likelihood function (Fig. 3d), the TP time and the acceptor fractions ($\varepsilon_1$ and $\varepsilon_2$) of the TP were determined for each transition. The bound,

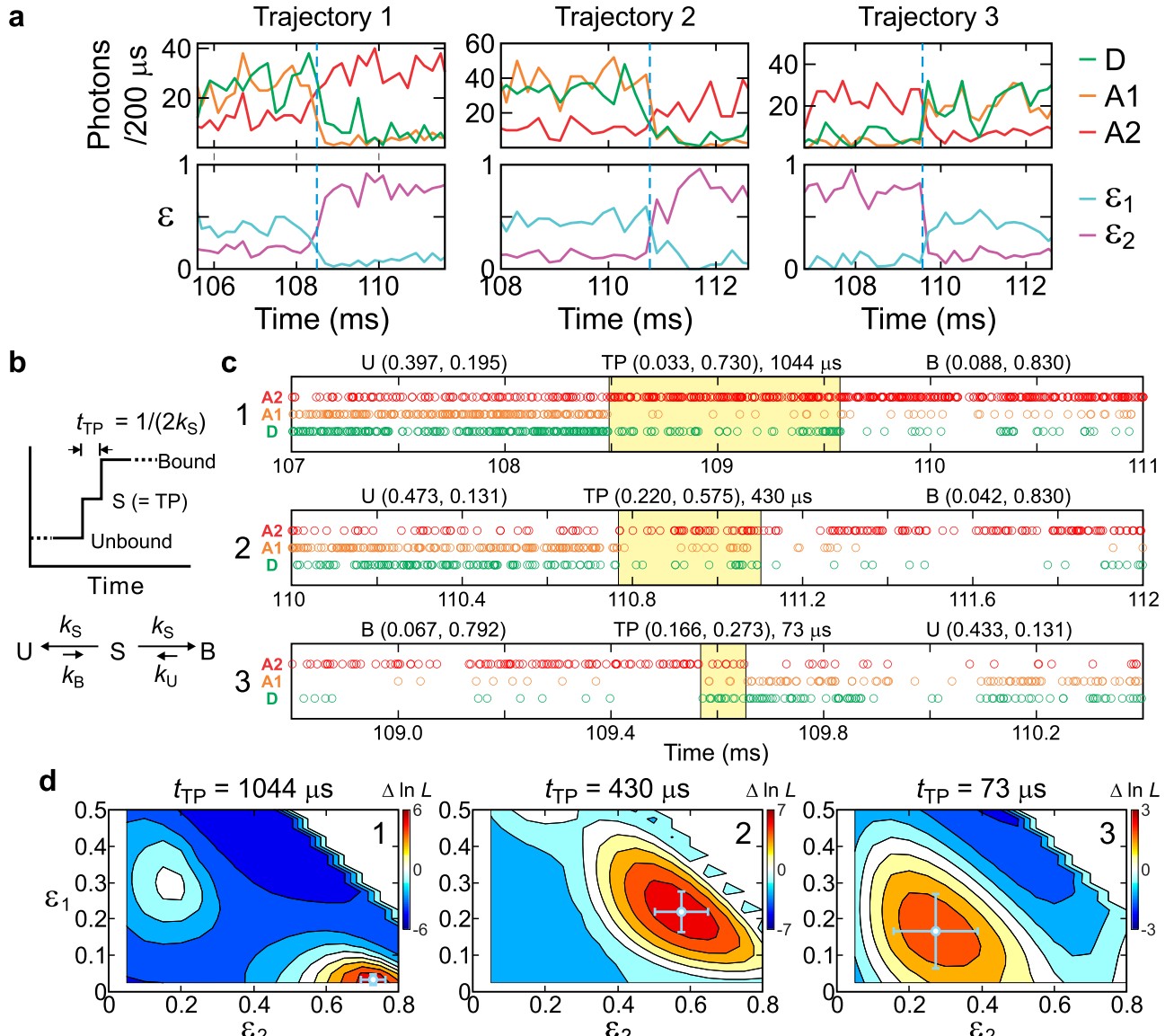

**Fig. 3 | Measurement of binding transition paths of TAD and Taz2.**
**a** Representative binned (200 μs bin time) fluorescence, $\varepsilon_1$, and $\varepsilon_2$ trajectories of high illumination intensity three-color FRET experiments of TAD-Taz2 binding. **b** A one-intermediate model for the maximum likelihood analysis of photon trajectories of individual transitions (see *Maximum likelihood method for the transition path analysis* in "Methods"). The TP is represented by the intermediate state (S) and its lifetime ($1/(2k_S)$) corresponds to the transition path time, $t_{TP}$. **c** Photon trajectories near the transitions of the three trajectories in (**a**). Photon trajectories are separated into the unbound state, TP (yellow shaded region), and the bound state using the Viterbi algorithm[28,53] and the $t_{TP}$ and acceptor fraction values determined from the maximum likelihood analysis (see *Maximum likelihood method for the*

*transition path analysis* in "Methods"). The optimized acceptor fractions ($\varepsilon_1$, $\varepsilon_2$) of the three states and $t_{TP}$ are listed above each trajectory. **d** The 2D likelihood plot as a function of $\varepsilon_1$ and $\varepsilon_2$ at the optimized $t_{TP}$ for the three transitions in (**c**). $\Delta \ln L$ (= ln $L(\tau_S) - L(0)$) is the difference of the log likelihood between the one-intermediate model and the instantaneous transition model ($t_{TP} = 0$). The log likelihood difference in the white area outside the likelihood peaks is smaller than the minimum value of the color bar. Error bars indicate standard deviations (SDs) obtained from the curvature at the maximum of the likelihood function. The TPs were measured at [NaCl] = 50 mM to expedite data collection because the binding kinetics is too slow at [NaCl] = 15 mM. See Supplementary Fig. 4 for the data collected at [NaCl] = 15 mM.

TP, and unbound regions determined from these parameters are indicated in the photon trajectories (Fig. 3c). The photon color pattern of the TP region differs from those of the bound and unbound states. In transition #1, for example, the A2 (red) photon count rate is very high in the bound state (right) and low in the unbound state (left). The D (green) count rate is lowest in the bound state and highest in the unbound state. These count rates show intermediate levels in the TP region (yellow shaded region in the middle). The pattern changes and the duration of TP vary across different transitions (Fig. 3c, d). In other words, TPs are heterogeneous.

The distribution of the parameters of individual transitions is shown in the two-dimensional $\varepsilon_1$ and $\varepsilon_2$ plot in Fig. 4a. The individual TP

times are indicated by different colors. The broad distribution in Fig. 4a can be clustered into two groups: one with long TP times localized in the lower right corner (TP2) with high $\varepsilon_2$ and low $\varepsilon_1$, and the other with a wide-spread distribution with relatively shorter TP times (TP1). The diversity of TP1 is much higher than that of TP2, similar to the binding TPs of TAD-NCBD binding[28]. To accurately determine the average parameters of these two TPs, the entire data were fitted to a two parallel TP model (double-TP model). This resulted in the TP times of 90 μs and 1.5 ms for TP1 and TP2, respectively, and the fraction of TP1 of 0.43 (Supplementary Table 4).

We note that these two TPs of TAD-Taz2 binding are indeed analogous to the two exchange pathways of Taz2 and Mdm2 in Fig. 2. $\varepsilon_1$

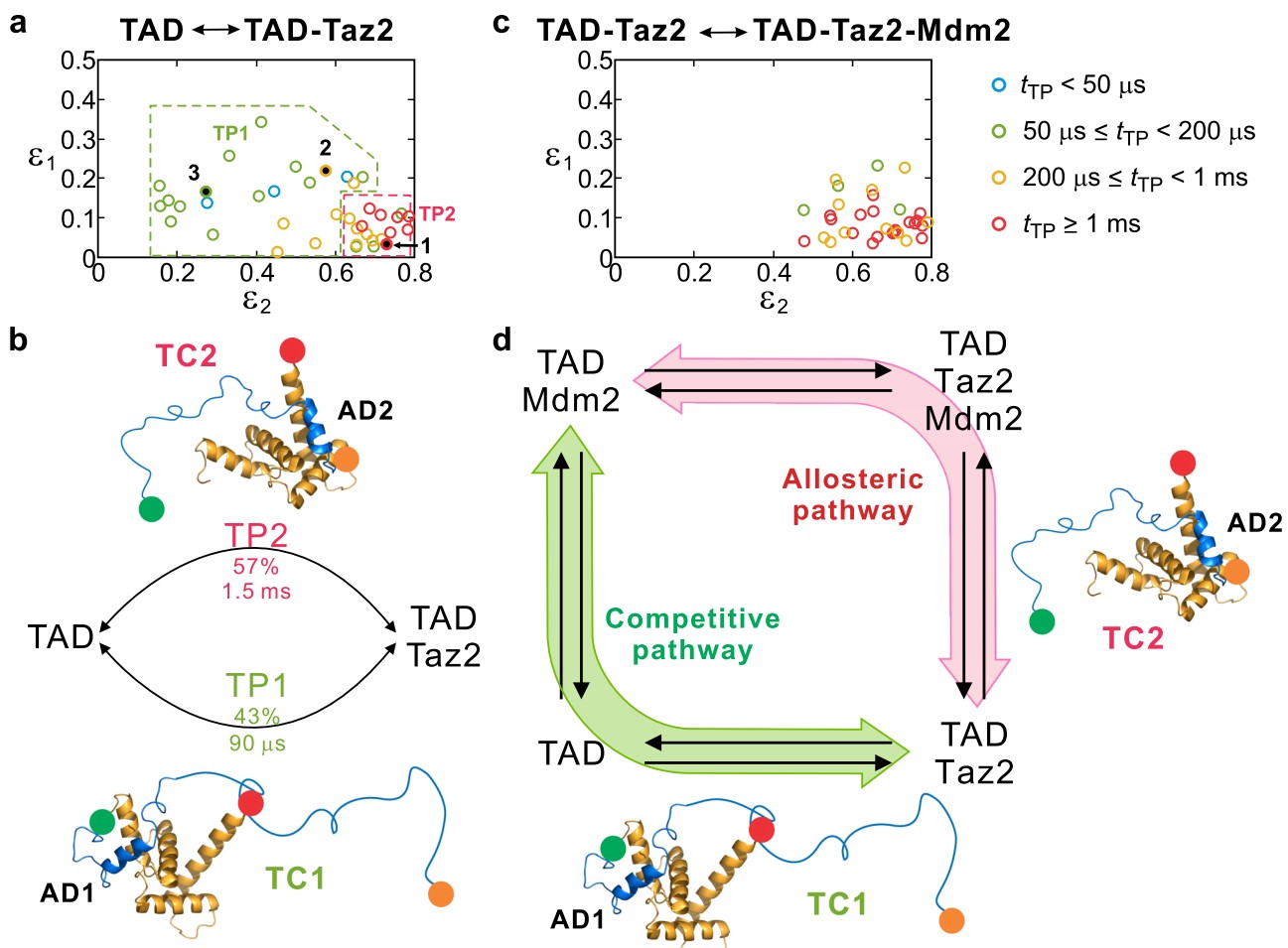

**Fig. 4 | Comparison of transition paths of TAD binding with Mdm2 and Taz2.**
**a** The distribution of $\varepsilon_1$ and $\varepsilon_2$ values of individual TPs of TAD and Taz2 binding in Fig. 3. Three black dots correspond to the three transitions shown in Fig. 3. The color of each data point indicates the range of $t_{TP}$. The distribution was clustered into two groups (TP1 and TP2) according to $\varepsilon_1$, $\varepsilon_2$, and $t_{TP}$. 94 transitions were analyzed. Only data with $\Delta\ln L > 1$ are included in the plot. **b** Global fitting of the transition data with a double-TP model. The fraction and average $t_{TP}$ values of TP1 and TP2 are indicated (see Supplementary Table 4). The transient complexes

appearing in TP1 and TP2 are denoted as TC1 and TC2, in which Taz2 is bound to AD1 and AD2, respectively. **c** The distribution of $\varepsilon_1$ and $\varepsilon_2$ values of individual TPs between the ternary complex and the TAD-Taz2 complex. 102 transitions were analyzed. Only data with $\Delta\ln L > 1$ are included in the plot. **d** The transient complexes TC1 and TC2 appear in the allosteric and competitive binding pathways of the exchange of the binding partners, Taz2 and Mdm2. The TPs for the ternary complex formation were measured at [NaCl] = 15 mM. Source data for (**a**) and (**c**) are provided as a Source Data file.

( = 0.093, Supplementary Table 4) of TP2 is very low similar to the fully bound state (0.094), indicating that TP2 involves a transient binding complex with A1 close to A2 (TC2, upper conformation in Fig. 4b). The relatively lower $\varepsilon_2$ ( = 0.75) in TC2 compared to the fully bound state (0.82) indicates that the donor-labeled N-terminus is more extended. In other words, AD2 is bound to Taz2 and AD1 is free in this transient complex TC2. This suggests that TP2 is related to the allosteric pathway above, in which a transient complex with free AD1 and Taz2-bound AD2 is required for Mdm2 binding to form the ternary complex (Fig. 4d). On the other hand, TP1, which has a shorter TP time, may involve a transient complex, in which AD1 is bound to Taz2 and AD2 is free (TC1, lower conformation in Fig. 4b). Since Mdm2 cannot bind to TC1, TAD must dissociate completely from Taz2 to bind to Mdm2. Therefore, TP1 is related with the competitive pathway (Fig. 4d). TAD can also dissociate from TC2 completely, which contributes to the competitive pathway, but this fraction is relatively small when the Mdm2 concentration is high.

If this hypothesis is true, the TP time of TP2 of TAD-Taz2 binding should be comparable to the TP time along the allosteric pathway (i.e., transitions between the ternary complex and the TAD-Taz2 complex) because both processes should involve the same AD2-bound and AD1-

free transient complex TC2. We performed an experiment on the formation of the ternary complex shown in Fig. 2e at a high illumination intensity and analyzed the transitions between the TAD-Taz2 complex and the ternary complex (see Supplementary Fig. 5 for example photon trajectories and likelihood plots). Indeed, the 2D plot in Fig. 4c shows that the TPs are localized in a region similar to the TP2 cluster of the TAD-Taz2 binding in Fig. 4a. The broader distribution than the TP2 boundary shown in Fig. 4a is due to statistical fluctuations. Fitting all transitions to the double-TP model results in similar transition path times of 0.8 ms and 1.15 ms with similar acceptor fractions (Supplementary Table 4), indicating the single-TP model is sufficient to describe this distribution. The acceptor fractions determined from a single-TP analysis, $(\varepsilon_1, \varepsilon_2) = (0.077, 0.721)$, are very close to those of TP2 of TAD-Taz2 binding, $(\varepsilon_{1TP2}, \varepsilon_{2TP2}) = (0.093, 0.747)$, and the average TP time of 921 μs is comparable to $t_{TP2}$, 1.5 ms (Supplementary Table 4). This similarity confirms that the allosteric pathway of Taz2 and Mdm2 exchange and TP2 of TAD-Taz2 binding share the same transient complex conformation, TC2. The other TP of TAD-Taz2 binding, TP1, is related to the competitive pathway. These results strongly support the hypothesis that the double pathways of binding partner exchange (Taz2 and Mdm2) are closely associated with the

diverse binding transition paths of TAD-Taz2 binding, originating from TAD's nature of multi-site interactions.

## Discussion

The transition path time of folding of natural proteins is extremely short on the order of 1 μs, regardless of the folding kinetics[29–31]. This is presumably the result of natural evolution toward avoiding local traps and non-native interactions, which smooths the folding energy landscape[32–34]. On the other hand, the TP time of IDP binding is two orders of magnitude longer[23,35]. One cause of this long TP times of IDP binding is the formation of transient binding complexes by non-native interactions[23]. Utilizing non-native interactions seems to be the opposite direction of the evolution of protein folding, but there are advantages. It has been shown that non-native interactions can actually facilitate binding[23] because binding can be initiated from different parts of an IDP especially when there are multiple binding sites. This results in heterogeneous binding pathways, as observed in the binding experiment of TAD and NCBD (another domain in CBP) using fast three-color FRET[28]. The same appears true for TAD-Taz2 binding in this work. The association rate constant ($3.5 \times 10^7 M^{-1} s^{-1}$) is lower than that of TAD-NCBD binding, but is still highly facilitated, and the binding TPs are similarly heterogeneous (Fig. 4a). Another advantage lies in IDPs' promiscuity. Heterogenous binding TPs offer more efficient pathways for binding partner exchange via ternary complex formation as demonstrated in this work.

The ability of IDPs to interact simultaneously with multiple binding partners allows for more efficient control of binding networks via allosteric modulation of binding characteristics, compared to individual two-component interactions[8,9]. This is particularly effective when there are multiple interaction sites, as is the case of the TAD. Ferreon et al.[36] showed that the adenovirus early region 1 A (E1A), an IDP, exhibits complex interactions with Taz2 and the retinoblastoma protein (pRb). Both Taz2 and pRb have two interaction sites in E1A and either positive or negative cooperativity was observed depending on the availability of the E1A interaction sites. In this case, regulating the availability of interaction sites optimizes component populations.

Modulations of binding affinity and kinetics via the formation of a ternary complex have also been observed in several other systems. The exchange of two IDPs, HIF-1α and its negative regulator CITED2, for their binding partner Taz1 of CBP was not symmetric, but unidirectional; CITED2 facilitated dissociation of HIF-1α[37]. This phenomenon was explained by the competitive interaction between the common motif of the two IDPs that interact with Taz1 in the ternary complex[37,38]. This allows for the efficient negative control of transcription at low concentrations of CITED2. Schuler and coworkers[39] showed that the dissociation of the positively charged linker histone H1.0 from its chaperone, negatively charged prothymosin α (ProTα), speeds up as ProTα concentration increases. This process was explained by the formation of a ternary complex consisting of one H1.0 and two ProTα followed by the dissociation of the H1.0-ProTα complex from the ProTα molecule that was originally bound to H1.0. This type of process is called competitive substitution[40]. The same principle explains the enhanced dissociation of tightly bound H1.0 from a nucleosome on a biologically relevant time scale through the formation of a ternary complex of nucleosome-H1.0-ProTα and the dissociation of the H1.0-ProTα complex[41].

Related with these previous studies, our study reveals another intricate yet efficient way of regulation involving diverse binding partner exchange pathways: a competitive pathway, where one protein completely dissociates and another protein associates, and an allosteric pathway, where exchange occurs through the formation of a ternary complex. Furthermore, our high time-resolution three-color single-molecule fluorescence experiments reveal the structural mechanism of this heterogeneity; the diversity in exchange pathways is encoded in the heterogeneous transition path of TAD and Taz2

binding, which is possible because Taz2 binds both AD1 and AD2 binding motifs in TAD while Mdm2 binds only to AD1.

Our three-color single-molecule measurements allow us to determine all kinetic rate constants along the pathways (Fig. 2i, Supplementary Table 2, see *Determination of rate constants* in "Methods"). Using these, we can quantitatively characterize the allosteric effects involving the ternary complex. The kinetic analysis shows that the allosteric effect is largest when Mdm2 binds in the presence of Taz2. Compared to the modest changes in the rates of all other processes (see *Analysis of allosteric effects* in "Methods" including Supplementary Fig. 6 for further allostery analyses), the association of Mdm2 is greatly reduced by the presence of Taz2, by a factor of 40 (from $4.2 \times 10^7 M^{-1} s^{-1}$ to $1.1 \times 10^6 M^{-1} s^{-1}$). There are two factors: (1) AD1 binding to Taz2 simply blocks the Mdm2 binding site and (2) even when AD1 detaches, Taz2-AD2 in vicinity can interfere with Mdm2 binding. The latter effect can result from steric hindrance because AD1 in TC2 is less free than in the free TAD, which is extended due to electrostatic repulsion[23,28]. Overall, this selective blockage reduces the Mdm2-bound TAD population and increases the Taz2-bound TAD population.

To assess the significance of the role of the allosteric pathway, we compared the flux of exchange of Mdm2 and Taz2 via the allosteric pathway and the competitive pathway, assuming that the physiological concentrations of CBP/p300 and Mdm2 are within a factor of ten of their respective dissociation constants (i.e., 10–1000%). (See *Crude estimation of concentrations of CBP/p300 and Mdm2 in the nucleus* in "Methods" for crude estimation of the physiological concentrations of CBP/p300 and Mdm2 of 3 nM and 460 nM, respectively.) The flux via the allosteric pathway increases with Mdm2 concentration from 40% at 200 nM to high values at μM concentrations, 76% and 86% at [Mdm2] = 1 and 2 μM, respectively (Supplementary Fig. 7a) (see *Calculation of the relative flux involving the ternary complex* in "Methods"). These calculated values are very close to the experimental values of 39% (200 nM) and 81% (2 μM) (see *Calculation of the relative flux involving the ternary complex* in "Methods" for the experimental determination). Therefore, the allosteric pathway enables not only the maintenance of the relatively high Taz2-bound complex that may prime activation of p53 through acetylation and phosphorylation, but also the rapid transfer of Mdm2-bound TAD to Taz2-bound TAD when needed. This allows for rapid switching of ligands with opposite functions (coactivator vs. negative regulator). The fraction along the allosteric pathway also depends on the affinity of Mdm2. It decreases as Mdm2 affinity decreases (Supplementary Fig. 7b). Mdm2 affinity is reduced by phosphorylation of S15, T18, and S20 in the TAD. For instance, phosphorylation at T18 reduces the affinity by 20-fold[13]. In this case, the fraction along the allosteric pathway decreases to 20% at [Mdm2] = 2 μM (Supplementary Fig. 7b).

As mentioned above, we previously observed heterogeneous TPs in TAD-NCBD binding[28] similar to those of TAD-Taz2 in this work. Since NCBD interacts with both AD1 and AD2 as does Taz2, these complex interaction dynamics may be a general feature of TAD binding when both binding sites interact with binding partners. Therefore, the effective binding partner exchange via an allosteric pathway and switching to a competitive pathway depending on relative affinity changes may also be a general control mechanism for binding networks of IDPs possessing multiple binding sites and heterogeneous binding TPs.

## Methods

### Protein expression, purification, and dye labeling

The preparation of TAD labeled with Alexa 488 and Alexa 594 can be found in Ref. 28.

The Taz2 domain (residue 1764–1855 of mouse CBP) was cloned (ATUM, Newark, CA) as a fusion protein with 6His-tag and GB1 linked by a thrombin cleavage site (LVPRGS). The 4 non-zinc-coordinating cysteines (16, 24, 67 and 68, Supplementary Fig. 1) were replaced with

alanine to obtain a stable and functional protein[42]. In addition, the amber codon (TAG) was introduced at the N-terminus to incorporate an unnatural amino acid[43]. Taz2 was expressed in *E. coli* BL21(DE3) cells that were co-transformed with the pEVOL-pAcF plasmid[43], which enables incorporation of an unnatural amino acid, 4-acetylphenylalanine, at the amber codon. Protein expression was induced at O.D. -0.7 at 600 nm by adding isopropyl β-D-1-thiogalactopyranoside (IPTG, final concentration of 1 mM), arabinose (final concentration of 1 mM), and 4-acetylphenylalanine hydrochloride (Synchem, #SC-35005) for 4 h at 37 °C. After harvesting the cells, the protein was purified using a column with IgG Sepharose (IgG Sepharose 6 Fast Flow affinity resin, Cytiva), which specifically binds GB1. GB1 was then removed by thrombin cleavage. Taz2 was labeled with CF680R hydroxylamine (Biotium, #92054) at the acetylphenylalanine residue by oxime ligation reaction[44].

Similarly, Mdm2 (residue 18 – 125, p53 binding domain) was cloned (ATUM, Newark, CA) and expressed as a GB1 fusion protein containing a thrombin cleavage site. This 6His-GB1-Mdm2 construct was purified using a His-tag affinity column, followed by size exclusion chromatography (Superdex 75 10/300 GL, Cytiva). 6His-GB1 was then cleaved from Mdm2 by thrombin and subsequently removed using a His-tag affinity column and size exclusion chromatography.

AD2 (residue 38–61 of human p53) was cloned (ATUM, Newark, CA) and expressed in a similar manner to TAD[28]. Briefly, the AD2 sequence was flanked by the AviTag sequence with a flexible linker (Supplementary Fig. 1 d). The construct was co-transformed into *E.coli* BL21 (DE3) cells with a pJ411-birA plasmid (ATUM, Newark, CA) to overexpress biotin ligase (BirA) for the biotinylation of AD2. AD2 was expressed as a GB1 fusion with a factor Xa cleavage site (IEGR). The cells were induced by adding 1 mM IPTG with 100 µM d-biotin at O.D - 0.7 at 600 nm. After harvesting the cells, the construct was purified using a His-tag affinity column and size exclusion chromatography. The product was additionally purified using a streptavidin mutein column (Roche, 03708152001) to ensure biotinylation. 6His-GB1 was then cleaved by factor Xa and subsequently removed using a His-tag affinity column and size exclusion chromatography. The C-terminal cysteine residue was labeled with Alexa 488 maleimide.

## Single-molecule fluorescence experiments

Single-molecule FRET experiments were performed using a confocal microscope system (MicroTime200, Picoquant) with a 75 µm diam. pinhole, a beamsplitter (Z488/594rpc, Chroma Technology), and an oil-immersion objective (UPLSAPO, NA 1.4, × 100, Olympus). Alexa 488 was excited by a 485 nm diode laser (LDH-D-C-485, PicoQuant) in the CW mode. Alexa 488, Alexa 594, and CF680R fluorescence was split into three channels using two beamsplitters (585DCXR and 670DCXR, Chroma Technology) and focused through optical filters (ET525/50 m for Alexa 488, ET645/75 m for Alexa 594, and ET705/72 m for CF680R, Chroma Technology) onto photon-counting avalanche photodiodes (SPCM-AQR-16, PerkinElmer Optoelectronics). The data were collected using SymPhoTime (v. 5, Picoquant) and MATLAB (2019b). The data were analyzed using MATLAB (2019b or 2023b) custom codes.

Biotinylated TAD molecules were immobilized on a biotin-embedded, polyethylene glycol-coated glass coverslip (Bio_01, Microsurfaces Inc.) via a biotin (surface)- NeutrAvidin-biotin (protein) linkage (Fig. 1b). The surface was initially incubated with NeutrAvidin (30 µg/mL) for 5 min and subsequently with TAD (80 pM) for 3 min. Then, immobilized TAD molecules were incubated with 17 nM Taz2 (TAD-Taz2 binding experiment), 200 nM Mdm2 (Mdm2-TAD binding experiment), or 2 µM (or 200 nM) Mdm2 and 17 nM Taz2 (ternary complex experiment). For transition path measurements, molecules were illuminated at 15 µW, which led to a photon count rate of ~300 ms⁻¹. All experiments were performed in 20 mM Tris-HCl buffer (pH 7) with 15 mM or 50 mM of NaCl. To reduce dye photobleaching

and blinking, 2 mM cyclooctatetraene (COT), 2 mM 4-nitrobenzyl alcohol (NBA), 2 mM trolox, 100 mM β-mercaptoethanol were used[45–47]. All experiments were performed at room temperature (22 °C).

## Determination of the acceptor 2 labeling efficiency of Taz2

The acceptor 2 (A2) labeling efficiency of Taz2 can be calculated using the number of transitions from the unbound TAD to either A2-labeled or unlabeled Taz2 in the TAD-Taz2 binding experiment, which were 235 and 33, respectively. These values result in an A2 labeling efficiency of 87.7%. The labeling efficiency can also be calculated in the ternary complex formation using the number of transitions from the TAD-Mdm2 complex to the TAD-Taz2 complex with or without active A2, which were 151 and 28, respectively, at [Mdm2] = 2 µM. The labeling efficiency of 84.4% is similar to those obtained from the binary TAD-Taz2 binding experiment above. The value obtained from the experiment at [Mdm2] = 200 nM was 93.9% (154 and 10 transitions, respectively). For consistency, the value of 87.7% was used in all calculations in this paper.

## Determination of rate constants

The rate constants in Fig. 2i were obtained from all three binding experiments involving TAD, Taz2, and Mdm2. Since there are 3 different bound complexes, with two binding partners, we first define the convention for the rate constants. For a reaction, $X + L \rightleftarrows C$, the association and dissociation rate constants are expressed as $k_{a,L}^C$ and $k_{d,L}^C$. C represents the bound complex: TAD-Taz2, TAD-Mdm2, and Ternary. L represents the binding partner that binds to form a complex C or dissociates from a complex C. For example, $k_{a,Mdm2}^{Ternary}$ and $k_{d,Mdm2}^{Ternary}$ are the association rate constant of Mdm2 and TAD-Taz2 to form the ternary complex and the dissociation rate constant of Mdm2 from the ternary complex, respectively. The binding and dissociation rates were determined from the corresponding waiting time distributions or the maximum likelihood method. The dissociation rate is equivalent to the dissociation rate constant. Binding is a pseudo-first-order reaction, and the association rate constant is obtained by dividing the binding rate by the concentration of the corresponding binding partner L.

The binding and dissociation rates of the TAD-Mdm2 binding were determined from the maximum likelihood analysis of photon trajectories (see the next section). Then, the association rate constant was obtained with [Mdm2] = 200 nM.

The binding and dissociation rates of the TAD-Taz2 binding were obtained by fitting the waiting time distributions in the unbound and bound states, respectively, to an exponential function as shown in Fig. 2g. The association rate constant was obtained with [Taz2] = 17 nM.

The rates involving the ternary complex along the allosteric pathway in Fig. 2i, were determined from the ternary complex formation experiment. The dissociation rates of Taz2 ($k_{d,Taz2}^{Ternary}$) and Mdm2 ($k_{d,Mdm2}^{Ternary}$) from the ternary complex were obtained from the waiting time distribution of the ternary complex (Fig. 2h) and the fractions of the number of transitions to the TAD-Mdm2 and TAD-Taz2 complexes (blue and purple dots in Fig. 2f, respectively), as

$$k_{d,Taz2}^{Ternary} = (\tau_{Ternary})^{-1} f_{Mdm2}$$
$$k_{d,Mdm2}^{Ternary} = (\tau_{Ternary})^{-1} (1 - f_{Mdm2}). \tag{1}$$

Here, $\tau_{Ternary}$ is the mean waiting time of the ternary complex obtained from the exponential fitting in Fig. 2h and $f_{Mdm2}$ is the fraction of the transition from the ternary complex to the TAD-Mdm2 complex (i.e., Taz2 dissociation). From the number of transitions from the ternary complex to TAD-Mdm2 and TAD-Taz2 complexes in Fig. 2f, which were 387 and 624, respectively, $f_{Mdm2} = 0.38$.

The binding rates to form the ternary complex from the TAD-Mdm2 and TAD-Taz2 complexes can be obtained from the waiting time distributions of the TAD-Mdm2 and TAD-Taz2 complexes, respectively, and the fractions of the transitions to the ternary complex. The association rate constants can then be obtained with [Mdm2] = 2 μM (or 200 nM) and [Taz2] = 17 nM.

Due to the accumulated error from three different experiments, the detailed balance is not satisfied in the kinetic scheme in Fig. 2i. In the calculation of the relative flux in the following section, we used a slightly reduced dissociation rate constant for TAD-Taz2 obtained by $k_{d,Taz2}^{TAD-Taz2} = k_{a,Taz2}^{TAD-Taz2} k_{a,Mdm2}^{Ternary} k_{d,Taz2}^{TAD-Taz2} k_{d,Mdm2}^{TAD-Mdm2} / (k_{d,Mdm2}^{Ternary} k_{a,Taz2}^{Ternary} k_{a,Mdm2}^{TAD-Mdm2}) = 0.14\,s^{-1}$ instead of $0.22\,s^{-1}$ obtained from the ternary complex formation experiment (Fig. 2g).

## Crude estimation of concentrations of CBP/p300 and Mdm2 in the nucleus

The physiological concentrations of CBP (Taz2) and Mdm2 were estimated based on their average copy number per cell (217 and 31,157, respectively)[48] and the diameter of the nucleus of 6 μm. This results in concentrations of 3 nM and 460 nM, respectively.

## Analysis of allosteric effects

As mentioned in the main text, the largest allosteric modulation is found in the association of Mdm2, which is reduced by the presence of Taz2 by a factor of 40 ($k_{a,Mdm2}^{Ternary} = 1.1 \times 10^6\,M^{-1}s^{-1}$ vs. $k_{a,Mdm2}^{TAD-Mdm2} = 4.2 \times 10^7\,M^{-1}s^{-1}$). The allosteric effects in other processes are relatively modest. Taz2 reduces the dissociation of Mdm2 by 72% ($k_{d,Mdm2}^{Ternary} = 2.2\,s^{-1}$ vs. $k_{d,Mdm2}^{TAD-Mdm2} = 7.9\,s^{-1}$). The apparent enhancement of the dissociation of Taz2 in the presence of Mdm2 is large, a factor of 10 ($3.6\,s^{-1}$ vs $0.38\,s^{-1}$). However, since Taz2 interacts only with AD2 in the presence of Mdm2, the proper comparison should be made with a construct containing only AD2 subdomain instead of the full-length TAD. We performed a binding experiment with the immobilized AD2 subdomain labeled with the donor and A2-labeled Taz2 (Supplementary Fig. 6). Both association and dissociation constants of Taz2 are affected only slightly by the presence of Mdm2 as $k_{a,Taz2}^{AD2-Taz2} = 4.8 \times 10^7\,M^{-1}s^{-1}$ vs. $k_{a,Taz2}^{Ternary} = 3.2 \times 10^7\,M^{-1}s^{-1}$ and $k_{d,Taz2}^{AD2-Taz2} = 1.5\,s^{-1}$ vs. $k_{d,Taz2}^{Ternary} = 2.2\,s^{-1}$.

## Calculation of the relative flux involving the ternary complex

The flux of the transition from the TAD-Mdm2 complex to the TAD-Taz2 complex in a non-equilibrium situation can be calculated using the steady-state approximation. The flux along the allosteric pathway via the ternary complex, $J_{allosteric}$ is calculated using $d[Ternary]/dt = k_{a,Taz2}^{Ternary}[TAD\text{-}Mdm2][Taz2] - (k_{d,Taz2}^{Ternary} + k_{d,Mdm2}^{Ternary})[Ternary] = 0$ as

$$J_{allosteric} = k_{d,Mdm2}^{Ternary}[Ternary]$$
$$= k_{a,Taz2}^{Ternary} \frac{k_{d,Mdm2}^{Ternary}}{k_{d,Taz2}^{Ternary} + k_{d,Mdm2}^{Ternary}} [Taz2][TAD-Mdm2]. \quad (2)$$

The flux of the transition along the competitive pathway, $J_{competitive}$ is calculated similarly using $d[TAD]/dt = k_{d,Mdm2}^{TAD-Mdm2}[TAD\text{-}Mdm2] - (k_{a,Mdm2}^{TAD-Mdm2}[Mdm2] + k_{a,Taz2}^{TAD-Taz2}[Taz2])[TAD] = 0$ as

$$J_{competitive} = k_{a,Taz2}^{TAD-Taz2}[TAD][Taz2]$$
$$= k_{d,Mdm2}^{TAD-Mdm2} \frac{k_{a,Taz2}^{TAD-Taz2}[Taz2]}{k_{a,Taz2}^{TAD-Taz2}[Taz2] + k_{a,Mdm2}^{TAD-Mdm2}[Mdm2]} [TAD-Mdm2].$$

$$(3)$$

Then, the fraction along the allosteric pathway can be obtained as

$$f_{allosteric} = \frac{J_{allosteric}}{J_{allosteric} + J_{competitive}}. \quad (4)$$

Experimentally, this fraction can be obtained from the number of transitions between the TAD-Mdm2 complex and TAD-Taz2 complex via the ternary complex and the number of direct transitions. These values were 199 and 107, respectively, at [Mdm2] = 2 μM, which results in $f_{allosteric} = 0.65$, lower than the calculated value of 0.86 (Supplementary Fig. 7a). This underestimation results from the short lifetime of the ternary complex of ~170 ms. 11% of the transition is shorter than the bin time 20 ms and cannot be resolved. These transitions are included in the competitive pathway. Therefore, the actual experimental value should be higher, $f_{allosteric} > 0.76$. The value of 0.29, calculated from 89 and 217 transitions, at a lower Mdm2 concentration of 200 nM was lower than that at [Mdm2] = 2 μM as expected from the curve in Supplementary Fig. 7a, but this value is also lower than the calculated value of 40%.

To obtain more accurate values, we analyzed photon trajectories directly, which is not affected by the bin time. It is possible to use the same likelihood function for the transition path analysis below (Eq. (7)). More specifically, all 306 transitions were analyzed together using a double-path model (see below Eqs. (13) and (14)). For each transition, up to 1000 photons from each side of the TAD-Taz2 and TAD-Mdm2 complexes and photons from the ternary complex were extracted and analyzed collectively. In the optimization, the lifetime of the intermediate state (i.e., free TAD) along the competitive pathway was fixed to the value expected at the Mdm2 concentration 2 μM, $1/k_{a,Mdm2}^{TAD-Mdm2}[Mdm2] = 11.8\,ms$ (See Supplementary Table 3 for the optimized parameters). The flux along the allosteric pathway determined this way is 0.39 and 0.81 at [Mdm2] = 200 nM and 2 μM, respectively, very close to the calculated value of 0.40 and 0.86 (Supplementary Fig. 7a). This analysis also determines the acceptor fractions of the intermediate (11.8 ms lifetime) between the TAD-Mdm2 and ternary complexes. The FRET efficiency (sum of acceptor 1 and acceptor 2 fractions, $\varepsilon_1^{competitive} + \varepsilon_2^{competitive}$ in Supplementary Table 3) of the intermediate along the competitive pathway, 0.48, is very close to that of free TAD, 0.46, verifying that the competitive pathway indeed involves free TAD.

## Determination of binding kinetics of TAD-Mdm2 using the maximum likelihood method

To determine the FRET efficiency and the two-state kinetic parameters of TAD-Mdm2 binding, we used the maximum likelihood method developed by Gopich and Szabo that analyzes photon trajectories directly without binning[24]. Since Mdm2 is unlabeled and there are only D and A1, most A2 photons result from the leakage of A1 photons. Therefore, A1 and A2 photons were treated as a single acceptor (i.e., two-color analysis). The likelihood function for the $j^{th}$ photon trajectory is

$$L_j = \mathbf{1}^\top \prod_{i=2}^{N_j} \left[ \mathbf{F}(c_i) \exp(\mathbf{K}\tau_i) \right] \mathbf{F}(c_1) \mathbf{p}_{eq}, \quad (5)$$

where $N_j$ is the number of photons in the $j^{th}$ trajectory, $c_i$ is the color of the $i^{th}$ photon (donor or acceptor), and $\tau_i$ is a time interval between the $(i-1)^{th}$ and $i^{th}$ photons. $\mathbf{K}$ is the rate matrix, the photon color matrix $\mathbf{F}$ depends on color $c$ of a photon as $\mathbf{F}(acceptor) = \mathbf{E}$ and $\mathbf{F}(donor) = \mathbf{I} - \mathbf{E}$, where $\mathbf{E}$ is a diagonal matrix with the apparent FRET efficiencies (the fraction of the combined A1 and A2 photons) of the individual states on the diagonal, $\mathbf{I}$ is the unity matrix, $\mathbf{1}^\top$ is the unit row vector (⊤ means transpose), and $\mathbf{p}_{eq}$ is the vector of equilibrium populations.

The likelihood function was calculated by the diagonalization of $\mathbf{K}$ as described in ref. [24]. Practically, the total log-likelihood function of all trajectories was calculated by summing individual log-likelihood functions as $\ln L = \sum_j \ln L_j$.

In this analysis, there are four fitting parameters: the apparent FRET efficiencies of the bound and unbound states, $E_B$ and $E_U$, the relaxation rate $k$, and the bound population $p_B$. The binding and dissociation rates are obtained as $k_B = k p_B$ and $k_U = k(1 - p_B)$. The matrix of FRET efficiencies, the rate matrix, and the vector of the equilibrium populations are given by

$$\mathbf{E} = \begin{pmatrix} E_B & 0 \\ 0 & E_U \end{pmatrix}, \mathbf{K} = \begin{pmatrix} -k_U & k_B \\ k_U & -k_B \end{pmatrix}, \mathbf{p}_{eq} = \begin{pmatrix} p_B \\ 1 - p_B \end{pmatrix}, \quad (6)$$

**Maximum likelihood method for the transition path analysis**

To determine the acceptor fractions and transition path time of individual transitions, we analyzed photon trajectories collected at high illumination intensity using the maximum likelihood method with a one-intermediate model shown in Fig. 3b, similar to the analysis in our previous work for binding of TAD and NCBD[28]. The trajectories containing a single transition are identified and separated for the likelihood analysis. The likelihood function for the photon trajectory of the $j^{th}$ transition is slightly different from that in Eq. (5) as[28,29]

$$L_j = \mathbf{v}_{fin} \cdot \prod_{i=2}^{N_j} [\mathbf{F}(c_i) \exp(\mathbf{K}\tau_i)] \mathbf{F}(c_1) \mathbf{v}_{ini}. \quad (7)$$

Here, $\mathbf{v}_{ini}$ and $\mathbf{v}_{fin}$ are the state vectors at the beginning and end of the trajectory. We reduced the association and dissociation rates by a factor of 1000 to effectively eliminate the contribution from multiple transitions that are not resolvable, i.e., $k_B (= k_B^m/1000)$ and $k_U (= k_U^m/1000)$, where $k_B^m$ and $k_U^m$ are measured binding and dissociation rates at low illumination intensity. $\mathbf{F}(acceptor\ 1) = \mathbf{E}_1$, $\mathbf{F}(acceptor\ 2) = \mathbf{E}_2$, and $\mathbf{F}(donor) = \mathbf{I} - \mathbf{E}_1 - \mathbf{E}_2$, where $\mathbf{E}_1$ and $\mathbf{E}_2$ are the diagonal matrices with the fractions of acceptor 1 (A1) and acceptor 2 (A2) photons. The one-intermediate model consists of 12 states: three protein states (bound, TP, and unbound) multiplied by two photophysical states (bright and dark) of each acceptor. The associated matrices are[28].

$$\mathbf{E}_I = Diag(\varepsilon_{IBbb}, \varepsilon_{ITPbb}, \varepsilon_{IUbb}, \varepsilon_{IBbd}, \varepsilon_{ITPbd}, \varepsilon_{IUbd}, \varepsilon_{IBdb}, \varepsilon_{ITPdb}, \varepsilon_{IUdb}, \varepsilon_{IBdd}, \varepsilon_{ITPdd}, \varepsilon_{IUdd}), \quad I = 1, 2$$

$$\mathbf{K} = \quad (8)$$

Here, $\mathbf{v}_{ini} = [0\ 0\ p_{b1}p_{b2}\ 0\ 0\ p_{b1}(1 - p_{b2})\ 0\ 0\ (1 - p_{b1})p_{b2}\ 0\ 0\ (1 - p_{b1})(1 - p_{b2})]^T$ and $\mathbf{v}_{fin} = [1\ 0\ 0\ 1\ 0\ 0\ 1\ 0\ 0\ 1\ 0\ 0]^T$ for a binding transition and $\mathbf{v}_{ini} = [p_{b1}p_{b2}\ 0\ 0\ p_{b1}(1 - p_{b2})\ 0\ 0\ (1 - p_{b1})p_{b2}\ 0\ 0\ (1 - p_{b1})(1 - p_{b2})\ 0\ 0]^T$ and $\mathbf{v}_{fin} = [0\ 0\ 1\ 0\ 0\ 1\ 0\ 0\ 1\ 0\ 0\ 1]^T$ for dissociation. $k_{bI}$ and $k_{dI}$ are the rate coefficients of the transitions from the dark to the bright state of acceptor $I$ and vice versa. The subscripts of the acceptor fraction b and

d stand for the bright and dark state of the acceptors. For example, $\varepsilon_{1bd}$ is the fraction of A1 when A1 is in the bright state and A2 is in the dark state. The fractions of A1 and A2 are defined as

$$\varepsilon_1 = \frac{n_{A1}}{n_{A1} + n_{A2} + n_D}$$
$$\varepsilon_2 = \frac{n_{A2}}{n_{A1} + n_{A2} + n_D}, \quad (9)$$

where $n_{A1}$, $n_{A2}$, and $n_D$ are photon count rates detected in A1, A2, and donor channels.

The rate matrix in Eq. (8) can also be expressed in a simpler way,

$$\mathbf{K} = \mathbf{I}_4 \otimes \mathbf{K}_{1S} + \mathbf{K}_{b1} \otimes \mathbf{I}_6 + \mathbf{I}_2 \otimes \mathbf{K}_{b2} \otimes \mathbf{I}_3, \quad (10)$$

where

$$\mathbf{K}_{1S} = \begin{pmatrix} -k_U & k_S & 0 \\ k_U & -2k_S & k_B \\ 0 & k_S & -k_B \end{pmatrix}, \quad (11)$$

$$\mathbf{K}_{bI} = \begin{pmatrix} -k_{dI} & k_{bI} \\ k_{dI} & -k_{bI} \end{pmatrix}, \quad I = 1, 2, \quad (12)$$

and $\mathbf{I}_J$ is the identity matrix of size $J \times J$. The Kronecker product $\mathbf{A} \otimes \mathbf{B}$ is a block matrix where each element of $\mathbf{A}$, $a_{ij}$, is multiplied by the entire matrix $\mathbf{B}$.

In Eq. (8), there are 24 acceptor fractions, but unknown parameters can be reduced by relating the values in the dark states of acceptors with those of two-color trajectory values (donor and A1, DA1; donor and A2, DA2)[28,49]. For A1 bright and A2 dark state, $\varepsilon_{1bd} = \varepsilon^{DA1}(1 - \varepsilon_{d12})$ and $\varepsilon_{2bd} = \varepsilon^{DA1}\varepsilon_{d12}$, where $\varepsilon_{d12}$ is the fraction of A1 photons detected in the A2 channel, which can be determined from DA1 trajectories. For A1 dark and A2 bright state, $\varepsilon_{1db} = (1 - \varepsilon^{DA2})\varepsilon_{d1}$ and $\varepsilon_{2db} = \varepsilon^{DA2}$. Here, $\varepsilon^{DA1}$ and $\varepsilon^{DA2}$ values were determined with the three-color acceptor fractions by global fitting of the three-color, DA1, and DA2 segments. For both A1 and A2 dark state, $\varepsilon_{1dd} = \varepsilon_{d1}$ and $\varepsilon_{2dd} = \varepsilon_{d2}$. $\varepsilon_{d1}$ ($\varepsilon_{d2}$) is the fraction of photons detected in A1 (A2) channel after donor excitation when A1 and A2 are in the dark state, which can be pre-determined from the donor-only segments. In the analysis of the transitions between the TAD-Taz2 and ternary complexes, only three-color segments were analyzed. Therefore, the two-color acceptor fractions were predetermined separately from the corresponding two-color segments, except for $\varepsilon_{1TPbd}$, $\varepsilon_{2TPbd}$, $\varepsilon_{1TPdb}$, and $\varepsilon_{2TPdb}$, for which the same values of TP2 of the TAD-Taz2 binary complex experiment were used as an approximation.

To obtain the average acceptor fractions and TP time, the entire transition data were optimized by maximizing the total likelihood function, $\ln L = \sum_j \ln L_j$. In this case, transitions were analyzed with the single-path (one-intermediate model, transitions between the ternary complex and TAD-Taz2 complex) or double-path models (two intermediates, TAD-Taz2 binding transitions). The rate matrix of the double-path model is a $16 \times 16$ matrix, which can be obtained as[28]

$$\mathbf{K} = \mathbf{I}_4 \otimes \mathbf{K}_{2S} + \mathbf{K}_{b1} \otimes \mathbf{I}_8 + \mathbf{I}_2 \otimes \mathbf{K}_{b2} \otimes \mathbf{I}_4, \quad (13)$$

$$\mathbf{K}_{2S} = \begin{pmatrix} -k_U & k_{S1} & k_{S2} & 0 \\ p_{S1}k_U & -2k_{S1} & 0 & p_{S1}k_B \\ (1 - p_{S1})k_U & 0 & -2k_{S2} & (1 - p_{S1})k_B \\ 0 & k_{S1} & k_{S2} & -k_B \end{pmatrix}. \quad (14)$$

Here, $p_{S1}$ is the fraction of the transitions via TP1. The transition path times of TP1 and TP2 are $t_{TP1} = (2k_{S1})^{-1}$ and $t_{TP2} = (2k_{S2})^{-1}$,

respectively. It is straightforward to express the color matrices, and $\mathbf{v}_{ini}$ and $\mathbf{v}_{fin}$ similar to Eq. (8).

## Reporting summary
Further information on research design is available in the Nature Portfolio Reporting Summary linked to this article.

## Data availability
The experimental data generated in this study and analysis codes have been deposited in Zenodo (https://doi.org/10.5281/zenodo. 18509684)[50]. Materials used in this study are available upon request. Source Data are provided as a Source Data file. Source data are provided with this paper.

## Code availability
The source codes and compiled libraries and analysis routines using MATLAB for the transition path analysis, instructions for optimization using a multi-thread CPU calculation[51], and example data are available in the GitHub repository (https://github.com/hoisunglab/FRET_TransitionPath)[52].

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

## Acknowledgements

We thank I. V. Gopich, C. -J. Feng, S. P. Carney, and E. Song for numerous helpful discussions. This research was supported by the Intramural Research Program of the National Institute of Diabetes and Digestive and Kidney Diseases (NIDDK) within the National Institutes of Health (NIH). The contributions of the NIH authors are considered Works of the United States Government. The findings and conclusions presented in this paper are those of the authors and do not necessarily reflect the views of the NIH or the U.S. Department of Health and Human Services.

## Author contributions

J.-Y.K. and H.S.C. conceived research and wrote the manuscript. J.-Y.K. prepared samples, performed experiments, and analyzed the data.

## Funding

## Competing interests

The authors declare no competing interests.
