## [Transparent Peer Review file · Nature Communications]

Dynamic control of IDP interaction network via diverse binding pathways

Corresponding Author: Dr Hoi Sung Chung

Version 0:

Reviewer comments:

Reviewer #1

(Remarks to the Author)

In the manuscript "Dynamic control of IDP interaction network via diverse binding pathways," Kim and Chung investigated the binding of the transactivation domain (TAD) of p53 with interaction partners murine double minute 2 (Mdm2) and the transcriptional 41 adapter zinc-binding domain 2 (Taz2) of the CREB-binding protein (CBP) using surface-immobilized three-color single-molecule FRET spectroscopy to characterize binding pathways. The work presented in the manuscript is similar to a previous work by the same authors (ref. #22), where the same technique was used to study TAD binding to the nuclear coactivator binding domain (NCBD) of CREB. The authors' results show formation of binary (Mdm2-TAD and TAD-Taz2) and ternary (Mdm2-TAD-Taz2) complexes through Competitive and Allosteric pathways, and different transition paths for TAD-Taz2 binding.

The following are comments/questions for the authors to consider:

1. Does TAD bind Taz2 to form only binary complexes, with Taz2 binding AD1, AD2 or both, or can a ternary complex (i.e., Taz2-TAD-Taz2) also form with different Taz2 molecules binding different AD sites (perhaps accessible when using higher Taz2 concentrations)?
2. Is the use of the term Allosteric to describe one of the pathways valid given the following: a. AD1 is binding site for both Mdm2 and Taz2 (competitive binding); b. Taz2 binding to AD2 also involves AD1 in the binary complex; and, c. binding of a protein to two different ligands can be non-allosteric/independent ?
3. To validate/check the reported individual rates and the model, Kd values can be calculated from the individual rates and converted to free energies (Supplementary Table 1?), and because Fig. 2i shows a thermodynamic cycle, should the sum of the free energies be equal to zero (within error limits)?
4. See lines 151-152: To observe the free, disordered TAD, why not also use lower Mdm2 concentration then?
5. The authors' kinetic model seems to assume induced folding as mechanism (vs. conformational selection).
6. Wouldn't most molecules that bind two different ligands exhibit "heterogeneity" as described by how the term is being used in, for example, line 194?
7. The authors used 200 microsecond binning time. How does the data change when bin time is varied?
8. The transition paths observed (TP1 or TP2) correspond to TAD AD1- or AD2-bound Taz2 (TC1 or TC2), respectively (Fig.4). Did the results also detect transition paths corresponding to binary TAD-Taz2 complex with both AD sites occupied by one Taz2 molecule (Fig.1 TAD-Taz2 structure cartoon)? Any evidence for Taz2-TAD-Taz2 ternary complex (or [Taz2] may be too low)?

Reviewer #2

(Remarks to the Author)

In this work, Kim and Chung use single-molecule FRET experiments and kinetic modeling to examine aspects of p53 binding specificity, cooperativity, and allostery. Specifically, they look at the intermolecular interactions of the transactivation domain (TAD) of p53 with the transcriptional adapter zinc-binding domain 2 (Taz2) of CBP and the E3 ubiquitin ligase murine double minute 2 (Mdm2) protein. TAD contains two binding sites for Taz2, one of which is also a binding site for Mdm2, permitting both binary and ternary complex formation. The authors hypothesize that binding and exchange of TAD ligands may be more complex than a simple competitive binding scheme, making the system a useful model to study multi-binding-site interactions. Using three colored FRET experiments and maximum likelihood analyses, the authors propose a complex, heterogeneous binding scheme where the three proteins interact via two pathways. The first pathway yields a binary complex, where dissociation of one TAD ligand is required for the binding of the other. The second pathway, driven by allostery, involves the formation of the ternary complex. Kinetic analyses of both pathways illustrate exchange of Mdm2 and Taz2 occurs on a faster timescale from the allosteric pathway as opposed to the simple competitive dissociation pathway. The experiments and analyses are technically challenging and well executed. However, the reviewer believes that the significance of this work would be greatly increased if additional experimental work would be conducted at multiple concentrations of Taz2 and Mdm2 to test predictions generated from the fractional flux models of Taz2 and Mdm2 exchange. Linking this data with a discussion on the cellular concentrations (if known) of Mdm2 and Taz2 under different physiological states would help provide credence to the functional relevance of formation of the ternary complex and the proposed allosteric pathway being important for binding exchange.

Major concerns and comments

- In the abstract, the authors propose that their work demonstrates how binding allostery can enable a faster response to changes in the external environment. Additionally, in the introduction, it is mentioned that under “stress situations” Mdm2 becomes replaced by other proteins to activate a distinct set of stress response genes, however, no references are included to support this idea. To provide physiological significance to this work regarding signaling under non-stressed and stressed conditions, a discussion should be included of how CBP and Mdm2 cellular concentrations 1) are altered under specific stress conditions, 2) can be related to the reported affinities and concentrations used in this work, and 3) would affect flux through the allosteric binding pathway analogous to the approach in Supplementary Figure 5 and the Discussion used to illustrate the effects of TAD phosphorylation on Mdm2 affinity.
- It would be useful to explicitly the way in which the authors define the pathways as either allosteric and competitive... both pathway leads to the formation of a ternary complex and it is unclear whether allostery plays a role on either pathways. Maybe I missed an important point in the definition of the system.
- There is no mention why the 21 amino acid linker was added to both TAD constructs (AD1/AD2 and AD2 only). Presumably this is to ensure the ADs are accessible, creating space from the surface attachment biotinylation site? Have the authors shown that surface attachment and addition of the linker do not affect binding to either Taz2 and Mdm2 alone and when added together?
- It is unclear how and under what conditions prior characterization and affinity measurements were made. Adding some additional details would provide context for the solution conditions and concentrations used here to help the reader of the current work.
- The transition path analysis seems to suggest that Taz2 binding induces distinct conformation changes in TAD, whether it is bound to AD1 or AD2. However, that is inferred from the transition pathways. It has no bearing on the final configuration of the complex, which is likely bind to both AD1 and AD2. So, the way in which I understand Fig. 4 is that there are two pathways to form the final complex, one that starts by binding AD1 and then form the complex with AD2 or viceversa. Binding of Mdm2 can happen (I would say not surprisingly) only by releasing AD1. It is great that this can be captured in the single-molecule data. Is there any information on other transition path times, e.g. when realizing Taz2 from the ternary complex?
- It would have been also interesting if the authors would have changed labeled positions to test whether AD1 and AD2 are both bound in the complex or only a fraction has both bound.
- The binding model appears to be dependent both on Taz2 and Mdm2 concentrations. Specifically, at Taz2 concentrations below 60 nM with corresponding Mdm2 concentrations below 0.5 μ M, a large variance in the fractional flux of exchange is observed relative to conditions where Mdm2 concentration is in excess (Supplementary Figure 5a), such as the 2 μ M tested here. The authors should perform experiments under these lower concentrations to directly test the model. Additionally, as the competitive dissociation model involves free (unbound) TAD, using lower Mdm2 concentrations should allow for direct detection of the unbound state.
- The manuscript would benefit of more examples of curves and availability of code used for data analysis.

Minor:

- In Supplementary Figure 1, color code sequences that correspond to AD1 and AD2. It is unclear for the “full-length” TAD, if all the sequence not found in the AD2 construct corresponds to AD1. I would guess not as in Figure 1a, there is a long linker depicted in between each AD. Plus, the title of the Figure refers to NCBBD, which is the protein that was studied in another manuscript.
- In Figure 1a, having AD1 and AD2 color coded the same as Mdm2 and Taz2, respectively is confusing. Would recommend to color code them as blue to be consistent with the rest of the figure. Polyethylene Glycol is misspelled on line 73.

Version 1:

Reviewer comments:

Reviewer #1

(Remarks to the Author)

The authors have thoroughly addressed my comments and questions, and the revised manuscript is substantially improved. I have no further questions or comments.

Reviewer #2

(Remarks to the Author)

I appreciate the patience and meticulous answer that the authors have put in addressing my concerns, where possible, and I have no further questions. This is an excellent work that demonstrates the power of single-molecule measurements in resolving kinetic pathways in IDP interactions.

Reviewer #1 (Remarks to the Author):

In the manuscript “Dynamic control of IDP interaction network via diverse binding pathways,” Kim and Chung investigated the binding of the transactivation domain (TAD) of p53 with interaction partners murine double minute 2 (Mdm2) and the transcriptional 41 adapter zinc-binding domain 2 (Taz2) of the CREB-binding protein (CBP) using surface-immobilized three-color single-molecule FRET spectroscopy to characterize binding pathways. The work presented in the manuscript is similar to a previous work by the same authors (ref. #22), where the same technique was used to study TAD binding to the nuclear coactivator binding domain (NCBD) of CREB. The authors’ results show formation of binary (Mdm2-TAD and TAD-Taz2) and ternary (Mdm2-TAD-Taz2) complexes through Competitive and Allosteric pathways, and different transition paths for TAD-Taz2 binding.

The following are comments/questions for the authors to consider:

→ We thank the reviewer for careful reading of the manuscript and thoughtful comments and suggestions. We revised the manuscript to address the reviewer’s comments, which strengthens our manuscript. Please see our responses below. Changes in the text are indicated in red.

After submission of the manuscript, we discovered that part of our analysis codes was lost during the upgrade of a computer. Since the raw data were still present, we could reanalyze. Although the conclusion remains unchanged, please note that there are small changes in the kinetic parameters. We believe this resulted from the difference in the inclusion or exclusion of noisy trajectories, which could not be reproduced. We will share all data and analysis codes that exactly produce figures and tables in the manuscript once it is accepted.

In addition, we omitted the correction of the Taz2 concentration due to incomplete labeling. We determined the acceptor 2 labeling efficiency to be 88% (*Determination of the acceptor 2 labeling efficiency of Taz2* in Methods), which increased the effective Taz2 concentration from 15 nM to 17 nM and slightly changed the rate constants.

While preparing the data set and analysis to be shared, we found some inconsistency in the parameters used in the analysis of the transition path of ternary complex formation. Fixing this issue resulted in changes to Fig. 4c and trajectory #2 in Supplementary Fig. 5. A brief discussion on Fig. 4c was added on page 10 (see below) and an associated double-TP analysis result was added to Supplementary Table 4. These changes didn’t alter any conclusions.

“Indeed, the 2D plot in Fig. 4c shows that the TPs are localized in a region similar to the TP2 cluster of the TAD-Taz2 binding in Fig. 4a. The broader distribution than the TP2 boundary shown in Fig. 4a is due to statistical fluctuations. Fitting all transitions to the double-TP model results in similar transition path times of 0.8 ms and 1.15 ms with similar acceptor fractions (Supplementary Table 4), indicating the single-TP model is sufficient to describe this distribution.”

Finally, we also performed an additional ternary complex formation experiment at a lower Mdm2 concentration of 200 nM as suggested. The results confirm our conclusion.

1. Does TAD bind Taz2 to form only binary complexes, with Taz2 binding AD1, AD2 or both, or can a ternary complex (i.e., Taz2-TAD-Taz2) also form with different Taz2 molecules binding different AD sites (perhaps accessible when using higher Taz2 concentrations)?

→ Binding of two Taz2 molecules to TAD will separate the donor and the acceptors, which would result in a state with distinct acceptor 1 and acceptor 2 fraction values as observed in the TAD-Taz2-Mdm2 ternary complex experiment. We did not observe a third state in the binding experiment of TAD and Taz2, indicating Taz2-TAD-Taz2 ternary complex does not exist at a Taz2 concentration of 17 nM.

The dissociation constant of the AD1 peptide is 27 μM (Ferreon et al. PNAS (2009) 106, 6591), which is 500 times higher than that of AD2, but it seems possible to form Taz2-TAD-Taz2 ternary complex at a higher concentration. However, once AD2 of the full-length TAD binds to Taz2, the effective concentration of intramolecular AD1 would be on the order of millimolar. Therefore, a ternary complex may be observable at a millimolar concentration of Taz2.

2. Is the use of the term Allostery to describe one of the pathways valid given the following: a. AD1 is binding site for both Mdm2 and Taz2 (competitive binding); b. Taz2 binding to AD2 also involves AD1 in the binary complex; and, c. binding of a protein to two different ligands can be non-allosteric/independent ?

→ We agree that a and b are correct. C is unclear to us. We believe that the binding of AD1 to Taz2 and Mdm2 is non-allosteric, but the presence of AD2 makes it allosteric. The measured allosteric effects are described in Discussion (page 12).

“Our three-color single-molecule measurements allow us to determine all kinetic rate constants along the pathways (Fig. 2i, Supplementary Table 2, see Determination of rate constants in Methods). Using these, we can quantitatively characterize the allosteric effects involving the ternary complex. The kinetic analysis shows that the allosteric effect is largest when Mdm2 binds in the presence of Taz2. Compared to the modest changes in the rates of all other processes (see Analysis of allosteric effects in Methods including Supplementary Fig. 6 for further allostery analyses), the association of Mdm2 is greatly reduced by the presence of Taz2, by a factor of 40 (from $4.2 \times 10^7 \text{ M}^{-1}\text{s}^{-1}$ to $1.1 \times 10^6 \text{ M}^{-1}\text{s}^{-1}$). There are two factors: (1) AD1 binding to Taz2 simply blocks the Mdm2 binding site and (2) even when AD1 detaches, Taz2-AD2 in vicinity can interfere with Mdm2 binding. The latter effect can result from steric hindrance because AD1 in TC2 is less free than in the free TAD, which is extended due to electrostatic repulsion^{23,28}. Overall, this selective blockage reduces the Mdm2-bound TAD population and increases the Taz2-bound TAD population.”

3. To validate/check the reported individual rates and the model, Kd values can be calculated from the individual rates and converted to free energies (Supplementary Table 1?), and because Fig. 2i shows a thermodynamic cycle, should the sum of the free energies be equal to zero (within error limits)?

→ The sum of the free energies is close to zero, but not exactly. In Fig. 2i, the dissociation rate of TAD and Taz2 complex calculated from the other seven rates is 0.14 s^{-1} (listed in Supplementary Table 2 and mentioned in the note below the table and on page 16, see below), which differs from the measured value of 0.22^{-1} . Therefore, there is a 1.6-fold difference in this cycle. The difference likely results from

accumulated errors in the parameters obtained from three different experiments (errors in the rate constants listed in Supplementary Table 2 range from 2 to 6%). However, the difference is relatively small, and we don't believe it affects the main conclusion.

(page 16)

“Due to the accumulated error from three different experiments, the detailed balance is not satisfied in the kinetic scheme in Fig. 2i. In the calculation of the relative flux in the following section, we used a slightly reduced dissociation rate constant for TAD-Taz2 obtained by $k_{d,Taz2}^{TAD-Taz2} = k_{a,Mdm2}^{TAD-Mdm2} k_{a,Mdm2}^{Ternary} k_{d,Taz2}^{Ternary} / (k_{a,Taz2}^{TAD-Taz2} k_{a,Taz2}^{Ternary} k_{d,Mdm2}^{Ternary} k_{d,Mdm2}^{TAD-Mdm2}) = 0.14 \text{ s}^{-1}$ instead of 0.22 s^{-1} obtained from the ternary complex formation experiment (Fig. 2g).”

4. See lines 151-152: To observe the free, disordered TAD, why not also use lower Mdm2 concentration then?

→ The difference in FRET efficiency between the Mdm2-bound TAD and free TAD is small (0.62 vs 0.46) and clear separation of these two states in binned trajectories is not possible for many transitions as shown in Fig. 2c and Supplementary Fig. 2b. Therefore, we performed the ternary complex experiment at a high Mdm2 concentration to simplify the analysis by eliminating the free TAD state. We believe this was a clever choice and, this certainly does not affect the conclusion.

However, we agree that it is worth verifying the presence of free TAD in the competitive pathway. We performed a maximum likelihood analysis of photon trajectories near transitions between the TAD-Mdm2 and TAD-Taz2 complexes to obtain the accurate fraction of transitions along the allosteric pathway ($f_{allosteric}$). In this analysis, we could also determine the FRET efficiency of a potential intermediate state (~12 ms lifetime) along the competitive pathway. The FRET efficiency was 0.48, very close to 0.46 of free TAD. This verifies free TAD is involved in the competitive pathway. This new analysis result was added to the main text (page 5) and *Calculation of the relative flux involving the ternary complex* section in Methods (page 17) and Supplementary Table 3.

(page 5)

“... This pathway involves free, disordered TAD, but it was not observed in *binned trajectories* due to the high Mdm2 concentration, as mentioned above. A maximum likelihood analysis of photon trajectories near the transitions determined that the FRET efficiency of a short segment (~12 ms) of the TAD-Mdm2 state right before the transition to TAD-Taz2 is 0.48, very close to that of free TAD in the TAD-Mdm2 binding experiment above (Fig. 2d). This result verifies that the competitive pathway involves free TAD. A similar result was obtained from the experiment with $[Mdm2] = 200 \text{ nM}$ (see *Calculation of the relative flux involving the ternary complex in Methods and Supplementary Table 3*).”

(page 17)

“... The flux along the allosteric pathway determined this way is 0.39 and 0.81 at $[Mdm2] = 200 \text{ nM}$ and $2 \text{ }\mu\text{M}$, respectively, very close to the calculated value of 0.40 and 0.86 (Supplementary Fig. 7a). This analysis also determines the acceptor fractions of the intermediate (~12 ms lifetime) between the TAD-Mdm2 and ternary complexes. The FRET efficiency (sum of acceptor 1 and acceptor 2 fractions, $\epsilon_1^{competitive} + \epsilon_2^{competitive}$ in Supplementary Table 3) of the intermediate along the competitive pathway,

0.48 (Supplementary Table 3), is very close to that of free TAD, 0.46, verifying that the competitive pathway indeed involves free TAD.”

In addition, we performed the experiment at a low Mdm2 concentration of 200 nM as suggested. The overall results are consistent with those of 2 μ M Mdm2 as summarized in Supplementary Fig. 3 and Table 2. However, free TAD was still not clearly observed in many transitions due to the similar FRET efficiency of free TAD and the TAD-Mdm2 complex as mentioned above.

5. The authors' kinetic model seems to assume induced folding as mechanism (vs. conformational selection).

→ The reviewer is correct. We are not aware of studies detecting pre-formed helices in TAD before binding to Taz2 or Mdm2. Even if such a helix exists, its lifetime would be extremely short, which is not observable in our measurement. Therefore, we assumed an induced folding model in the kinetic analysis.

6. Wouldn't most molecules that bind two different ligands exhibit "heterogeneity" as described by how the term is being used in, for example, line 194?

→ Binding would not be generally heterogeneous when an IDP can interact with two different ligands. If they share the binding sites entirely or partially and one ligand must dissociate completely for binding of the other ligand (i.e., competitive binding), then the ligand exchange pathway will be homogeneous. TAD-Taz2-Mdm2 is a unique system because TAD has two binding sites, and Taz2 interacts with both, while Mdm2 interacts with only AD1. In addition, TAD remains bound to Taz2 even when AD1 dissociates. These characteristics open up a new allosteric pathway and makes the ligand exchange pathway heterogeneous. One of the major discoveries of our study is the connection between heterogeneous ligand exchange pathways and the diverse binding pathways of TAD and Taz2.

7. The authors used 200 microsecond binning time. How does the data change when bin time is varied?

→ The trajectories collected at high illumination power were plotted with a 200 microsecond bin time as shown in Fig. 3a as the reviewer pointed out. However, this was only for visualization purposes. The actual transition path analysis was performed using photon trajectories without binning. Therefore, the bin time does not affect the analysis results. At low illumination power, the bin time was 20 ms as shown in Fig. 2. In this case, the bin time is much shorter than the waiting times, so the bin time has minimal effect on the results.

8. The transition paths observed (TP1 or TP2) correspond to TAD AD1- or AD2-bound Taz2 (TC1 or TC2), respectively (Fig.4). Did the results also detect transition paths corresponding to binary TAD-Taz2 complex with both AD sites occupied by one Taz2 molecule (Fig.1 TAD-Taz2 structure cartoon)? Any evidence for Taz2-TAD-Taz2 ternary complex (or [Taz2] may be too low)?

→ We are unsure if we understood the question correctly, but the binary TAD-Taz2 complex with both AD sites being occupied by one Taz2 molecule is the complete bound state (Fig. 1 cartoon, lower right

side). The two transition paths connecting this complex and the unbound TAD are TP1 (AD1 bound) and TP2 (AD2 bound). As described in the response to comment #1 above, we don't have evidence of the presence of Taz2-TAD-Taz2 ternary complex in our experimental condition.

Reviewer #2 (Remarks to the Author):

In this work, Kim and Chung use single-molecule FRET experiments and kinetic modeling to examine aspects of p53 binding specificity, cooperativity, and allostery. Specifically, they look at the intermolecular interactions of the transactivation domain (TAD) of p53 with the transcriptional adapter zinc-binding domain 2 (Taz2) of CBP and the E3 ubiquitin ligase murine double minute 2 (Mdm2) protein. TAD contains two binding sites for Taz2, one of which is also a binding site for Mdm2, permitting both binary and ternary complex formation. The authors hypothesize that binding and exchange of TAD ligands may be more complex than a simple competitive binding scheme, making the system a useful model to study multi-binding-site interactions. Using three colored FRET experiments and maximum likelihood analyses, the authors propose a complex, heterogenous binding scheme where the three proteins interact via two pathways. The first pathway yields a binary complex, where dissociation of one TAD ligand is required for the binding of the other. The second pathway, driven by allostery, involves the formation of the ternary complex. Kinetic analyses of both pathways illustrate exchange of Mdm2 and Taz2 occurs on a faster timescale from the allosteric pathway as opposed to the simple competitive dissociation pathway.

The experiments and analyses are technically challenging and well executed. However, the reviewer believes that the significance of this work would be greatly increased if additional experimental work would be conducted at multiple concentrations of Taz2 and Mdm2 to test predictions generated from the fractional flux models of Taz2 and Mdm2 exchange. Linking this data with a discussion on the cellular concentrations (if known) of Mdm2 and Taz2 under different physiological states would help provide credence to the functional relevance of formation of the ternary complex and the proposed allosteric pathway being important for binding exchange.

→ We thank the reviewer for careful reading of the manuscript and thoughtful comments and suggestions. We revised the manuscript to address the reviewer's comments, which we believe strengthens our manuscript. Please see our responses below. Changes in the text are indicated in red.

After submission of the manuscript, we discovered that part of our analysis codes was lost during the upgrade of a computer. Since the raw data were still present, we could reanalyze. Although the conclusion remains unchanged, please note that there are small changes in the kinetic parameters. We believe this resulted from the difference in the inclusion or exclusion of noisy trajectories, which could not be reproduced. We will share all data and analysis codes that exactly produce figures and tables in the manuscript once it is accepted.

In addition, we omitted the correction of the Taz2 concentration due to incomplete labeling. We determined the acceptor 2 labeling efficiency to be 88% (*Determination of the acceptor 2 labeling efficiency of Taz2* in Methods), which increased the effective Taz2 concentration from 15 nM to 17 nM and slightly changed the rate constants.

While preparing the data set and analysis to be shared, we found some inconsistency in the parameters used in the analysis of the transition path of ternary complex formation. Fixing this issue resulted in changes to Fig. 4c and trajectory #2 in Supplementary Fig. 5. A brief discussion on Fig. 4c was added on

page 10 (see below) and an associated double-TP analysis result was added to Supplementary Table 4. These changes didn't alter any conclusions.

“Indeed, the 2D plot in Fig. 4c shows that the TPs are localized in a region similar to the TP2 cluster of the TAD-Taz2 binding in Fig. 4a. The broader distribution than the TP2 boundary shown in Fig. 4a is due to statistical fluctuations. Fitting all transitions to the double-TP model results in similar transition path times of 0.8 ms and 1.15 ms with similar acceptor fractions (Supplementary Table 4), indicating the single-TP model is sufficient to describe this distribution.”

Finally, we also performed an additional ternary complex formation experiment at a lower Mdm2 concentration of 200 nM as suggested. The results confirm our conclusion.

Major concerns and comments

1. In the abstract, the authors propose that their work demonstrates how binding allostery can enable a faster response to changes in the external environment. Additionally, in the introduction, it is mentioned that under “stress situations” Mdm2 becomes replaced by other proteins to activate a distinct set of stress response genes, however, no references are included to support this idea.

→ We thank the reviewer for pointing this out. We amended the text to clarify the statement with relevant references on page 2.

“Under normal conditions, the TAD is bound to Mdm2, keeping the p53 level low¹⁴. On the other hand, under stressed situations, TAD is phosphorylated^{15,16}, and Mdm2 is replaced by other proteins to activate many stress response genes¹⁰. Phosphorylation of serine and threonine residues in TAD reduces the Mdm2 affinity^{13,17-19} and increase the affinity of the p300/CBP coactivator domains including Taz2^{13,19,20}.”

To provide physiological significance to this work regarding signaling under non-stressed and stressed conditions, a discussion should be included of how CBP and Mdm2 cellular concentrations 1) are altered under specific stress conditions, 2) can be related to the reported affinities and concentrations used in this work, and 3) would affect flux through the allosteric binding pathway analogous to the approach in Supplementary Figure 5 and the Discussion used to illustrate the effects of TAD phosphorylation on Mdm2 affinity.

→ To the best of our knowledge, the physiological concentrations of these proteins, especially how much they vary depending on stress conditions are unknown. We made a crude estimation using the copy number of proteins found in the literature on page 16. However, we don't know how accurate these values are.

(page 16)

“Crude estimation of concentrations of CBP/p300 and Mdm2 in the nucleus

The physiological concentrations of CBP (Taz2) and Mdm2 were estimated based on their average copy number per cell (217 and 31157, respectively)⁴⁷ and the diameter of the nucleus of 6 μm. This results in concentrations of 3 nM and 460 nM, respectively.”

The lack of knowledge on the accurate concentrations prevents more quantitative discussion. For this reason, we plotted the calculation of the dependencies over a wide range of concentrations in Supplementary Fig. 5 (Supplementary Fig. 7 in the revised manuscript). We hope the reviewer acknowledges the difficulty. We added several phrases related to this point on page 13.

“To assess the significance of the role of the allosteric pathway, we compared the flux of exchange of Mdm2 and Taz2 via the allosteric pathway and the competitive pathway, assuming that the physiological concentrations of CBP/p300 and Mdm2 are within a factor of ten of their respective dissociation constants (i.e., 10 – 1000%). (See Crude estimation of concentrations of CBP/p300 and Mdm2 in the nucleus in Methods for crude estimation of the physiological concentrations of CBP/p300 and Mdm2 of 3 nM and 460 nM, respectively.)”

2. It would be useful to explicit the way in which the authors define the pathways as either allosteric and competitive... both pathway leads to the formation of a ternary complex and it is unclear whether allostery plays a role on either pathways. Maybe I missed an important point in the definition of the system.

→ There is no allosteric effect in the competitive pathway and this pathway does not involve the ternary complex. The two pathways are defined on page 5, and we added clarification.

“One pathway for exchanging the binding partners involves the formation of the ternary complex (magenta pathway in Fig. 2i). We refer to this pathway as an allosteric pathway because two binding partners may affect each other’s binding (see Discussion). In the second pathway, the exchange of the binding partners occurs with complete dissociation of one partner followed by association of the other. This pathway does not involve the ternary complex and there is no allosteric effect. We refer to this pathway as a competitive pathway (green pathway in Fig. 2i).”

The allosteric effect in the allosteric pathway is described in Discussion (page 12).

“Our three-color single-molecule measurements allow us to determine all kinetic rate constants along the pathways (Fig. 2i, Supplementary Table 2, see Determination of rate constants in Methods). Using these, we can quantitatively characterize the allosteric effects involving the ternary complex. The kinetic analysis shows that the allosteric effect is largest when Mdm2 binds in the presence of Taz2. Compared to the modest changes in the rates of all other processes (see Analysis of allosteric effects in Methods including Supplementary Fig. 6 for further allostery analyses), the association of Mdm2 is greatly reduced by the presence of Taz2, by a factor of 40 (from $4.2 \times 10^7 \text{ M}^{-1}\text{s}^{-1}$ to $1.1 \times 10^6 \text{ M}^{-1}\text{s}^{-1}$). There are two factors: (1) AD1 binding to Taz2 simply blocks the Mdm2 binding site and (2) even when AD1 detaches, Taz2-AD2 in vicinity can interfere with Mdm2 binding. The latter effect can result from steric hindrance because AD1 in TC2 is less free than in the free TAD, which is extended due to electrostatic repulsion^{23,28}. Overall, this selective blockage reduces the Mdm2-bound TAD population and increases the Taz2-bound TAD population.”

We hope the above clarify the double pathway model of binding partner exchange.

3. There is no mention why the 21 amino acid linker was added to both TAD constructs (AD1/AD2 and AD2 only). Presumably this is to ensure the ADs are accessible, creating space from the surface attachment biotinylation site? Have the authors shown that surface attachment and addition of the linker do not affect binding to either Taz2 and Mdm2 alone and when added together?

→ Yes, we added a sentence to the legend of Supplementary Fig. 1 as

“Biotin is attached to the lysine residue (blue K) in the AviTag sequence, which is separated from TAD sequence by a flexible linker, providing space between the surface and the immobilized proteins and minimizing immobilization effect on binding.”

→ We added a table (Supplementary Table 1) that compares the dissociation constants measured in our study and previous studies. The similar values ensure minimal effects of immobilization and dye labeling. A sentence below was also added at the end of Introduction (page 2).

“The effect of immobilization and dye labeling seems minimal as the measured dissociation constants are similar to previously measured values^{12,13,17–19} (Supplementary Table 1).”

4. It is unclear how and under what conditions prior characterization and affinity measurements were made. Adding some additional details would provide context for the solution conditions and concentrations used here to help the reader of the current work.

→ We listed the solution conditions for measuring dissociation constants in various studies in Supplementary Table 1. The slightly lower dissociation constant of the TAD-Taz2 complex in our study may be due to the lower NaCl concentration compared to other studies.

5. The transition path analysis seems to suggest that Taz2 binding induces distinct conformation changes in TAD, whether it is bound to AD1 or AD2. However, that is inferred from the transition pathways. It has no bearing on the final configuration of the complex, which is likely bind to both AD1 and AD2.

→ The reviewer is correct. In the TAD-Taz2 bound complex, both AD1 and AD2 are bound as seen in the NMR structure. The conformations of the transient complexes during the transition path, which are AD1 bound and AD2 bound state, respectively, were deduced from the measured acceptor fractions during the transition paths (Fig. 4a). We believe this is a reasonable interpretation.

So, the way in which I understand Fig. 4 is that there are two pathways to form the final complex, one that starts by binding AD1 and then form the complex with AD2 or viceversa. Binding of Mdm2 can happen (I would say not surprisingly) only by releasing AD1. It is great that this can be captured in the single-molecule data. Is there any information on other transition path times, e.g. when realizing Taz2 from the ternary complex?

→ Yes, binding of Mdm2 occurs when AD1 is released, as measured in our experiment. The acceptor fractions, ϵ_1 and ϵ_2 , of the transition path between the ternary complex and TAD-Taz2 bound complex and their transition path times are plotted in Fig. 4c. This distribution is similar to that of TP2 (Fig. 4a)

represented by transient complex 2 (TC2) with AD1 being released. The results are discussed in the following paragraph (page 10).

*“If this hypothesis is true, the TP time of TP2 of TAD-Taz2 binding should be comparable to the TP time along the allosteric pathway (i.e., transitions between the ternary complex and the TAD-Taz2 complex) because both processes should involve the same AD2-bound and AD1-free transient complex TC2. We performed an experiment on the formation of the ternary complex shown in Fig. 2e at a high illumination intensity and analyzed the transitions between the TAD-Taz2 complex and the ternary complex (see Supplementary Fig. 5 for example photon trajectories and likelihood plots). Indeed, the 2D plot in Fig. 4c shows that the TPs are localized in a region similar to the TP2 cluster of the TAD-Taz2 binding in Fig. 4a. **The broader distribution than the TP2 boundary shown in Fig. 4a is due to statistical fluctuations. Fitting all transitions to the double-TP model results in similar transition path times of 0.8 ms and 1.15 ms with similar acceptor fractions (Supplementary Table 4), indicating the single-TP model is sufficient to describe this distribution. The acceptor fractions determined from a single-TP analysis, $(\epsilon_1, \epsilon_2) = (0.077, 0.721)$, are very close to those of TP2 of TAD-Taz2 binding, $(\epsilon_{1TP2}, \epsilon_{2TP2}) = (0.093, 0.747)$, and the average TP time of 921 μs is comparable to t_{TP2} , 1.5 ms (Supplementary Table 4). This similarity confirms that the allosteric pathway of Taz2 and Mdm2 exchange and TP2 of TAD-Taz2 binding share the same transient complex conformation, TC2. The other TP of TAD-Taz2 binding, TP1, is related to the competitive pathway. These results strongly support the hypothesis that the double pathways of binding partner exchange (Taz2 and Mdm2) are closely associated with the diverse binding transition paths of TAD-Taz2 binding, originating from TAD’s nature of multi-site interactions.”***

→ The transition path time of Taz2 binding to the TAD-Mdm2 complex to form the ternary complex or the dissociation of Taz2 from the ternary complex was too short to be measured. We did not include this information because it is not relevant to our discussion.

6. It would have been also interesting if the authors would have changed labeled positions to test whether AD1 and AD2 are both bound in the complex or only a fraction has both bound.

→ We chose the labeling positions that bring all three dyes close together when both AD1 and AD2 are bound to Taz2. The segments with very high acceptor 2 count rate and low donor and acceptor 1 count rates (indicated by red bars above the trajectories) in Fig. 2a correspond to this bound state. When one of AD1 or AD2 dissociates, eps1 and eps 2 will resemble those of TPs. Since the state with a high acceptor 2 count rate is the only bound state with an active acceptor 2 fluorophore, the fraction of the complex with only one of the ADs bound must be very low. TAD is a small molecule and donor and acceptor 1 are attached to its ends. The labeling positions are limited and changing them is unlikely to provide additional information.

7. The binding model appears to be dependent both on Taz2 and Mdm2 concentrations. Specifically, at Taz2 concentrations below 60 nM with corresponding Mdm2 concentrations below 0.5 μM , a large variance in the fractional flux of exchange is observed relative to conditions where Mdm2 concentration is in excess (Supplementary Figure 5a), such as the 2 μM tested here. The authors should perform experiments under these lower concentrations to directly test the model.

→ We mislabeled the curves in Supplementary Fig. 5 (Supplementary Fig. 7 in the revised manuscript). At 17 nM Taz2 (15 nM in the original manuscript), the variation of the flux is large enough to be tested. Therefore, we performed an experiment at $[Taz2] = 17$ nM and a lower concentration of Mdm2 of 200 nM. The representative trajectories and two-dimensional transition maps shown in Supplementary Fig. 3 indicate that the results are consistent with those measured at 2 μ M Mdm2. Kinetic analysis also yielded similar rate constants to those at 2 μ M Mdm2 as listed in Supplementary Table 2.

We obtained the fraction of transitions along the allosteric pathway at 200 nM and 2 μ M Mdm2 by counting the number of apparent transitions along the two pathways in binned trajectories. As shown in Supplementary Fig. 7 (blue dots), the values are lower than the calculated curve. This is due to an underestimation of the number of transitions along the allosteric pathway because the residence time in the ternary complex is sometimes shorter than the bin time 20 ms, which is not visible. For a more accurate estimation, we performed a maximum likelihood analysis of photon trajectories near transitions between the TAD-Mdm2 and TAD-Taz2 complexes. The values obtained in this way (red dots in Supplementary Fig. 7, parameters listed in Supplementary Table 3) agree very well with the calculated value, which verifies our model. The analysis and results are described on page 13,

“To assess the significance of the role of the allosteric pathway, we compared the flux of exchange of Mdm2 and Taz2 via the allosteric pathway and the competitive pathway, assuming that the physiological concentrations of CBP/p300 and Mdm2 are within a factor of ten of their respective dissociation constants (i.e., 10 – 1000%). (See Crude estimation of concentrations of CBP/p300 and Mdm2 in the nucleus in Methods for crude estimation of the physiological concentrations of CBP/p300 and Mdm2 of 3 nM and 460 nM, respectively.) The flux via the allosteric pathway increases with Mdm2 concentration from 40% at 200 nM to high values at μ M concentrations, 76% and 86% at $[Mdm2] = 1$ and 2 μ M, respectively (Supplementary Fig. 7a) (see Calculation of the relative flux involving the ternary complex in Methods). These calculated values are very close to the experimental values of 39% (200 nM) and 81% (2 μ M) (see Calculation of the relative flux involving the ternary complex in Methods for the experimental determination). Therefore, the allosteric pathway enables not only the maintenance of the relatively high Taz2-bound complex ...”

and in Calculation of the relative flux involving the ternary complex section in Methods (page 17)

“Experimentally, this fraction can be obtained from the number of transitions between the TAD-Mdm2 complex and TAD-Taz2 complex via the ternary complex and the number of direct transitions. These values were 199 and 107, respectively, at $[Mdm2] = 2$ μ M, which results in $f_{allosteric} = 0.65$, lower than the calculated value of 0.86 (Supplementary Fig. 7a). This underestimation results from the short lifetime of the ternary complex of ~ 170 ms. 11% of the transition is shorter than the bin time 20 ms and cannot be resolved. These transitions are included in the competitive pathway. Therefore, the actual experimental value should be higher, $f_{allosteric} > 0.76$. The value of 0.29, calculated from 89 and 217 transitions, at a lower Mdm2 concentration of 200 nM was lower than that at $[Mdm2] = 2$ μ M as expected from the curve in Supplementary Fig. 7a, but this value is also lower than the calculated value of 40%.

To obtain more accurate values, we analyzed photon trajectories directly, which is not affected by the bin time. It is possible to use the same likelihood function for the transition path analysis below (Eq. (7)). More specifically, all 306 transitions were analyzed together using a double-path model (see below Eqs. (13) and (14)). For each transition, up to 1000 photons from each side of the TAD-Taz2 and TAD-Mdm2

complexes and photons from the ternary complex were extracted and analyzed collectively. In the optimization, the lifetime of the intermediate state (i.e., free TAD) along the competitive pathway was fixed to the value expected at the Mdm2 concentration 2 μ M, $1/k_{a,Mdm2}^{TAD-Mdm2} [Mdm2] = 11.8$ ms (See Supplementary Table 3 for the optimized parameters). The flux along the allosteric pathway determined this way is 0.39 and 0.81 at $[Mdm2] = 200$ nM and 2 μ M, respectively, very close to the calculated value of 0.40 and 0.86 (Supplementary Fig. 7a). This analysis also determines the acceptor fractions of the intermediate (~ 12 ms lifetime) between the TAD-Mdm2 and ternary complexes. The FRET efficiency (sum of acceptor 1 and acceptor 2 fractions, $\epsilon_1^{competitive} + \epsilon_2^{competitive}$ in Supplementary Table 3) of the intermediate along the competitive pathway, 0.48 (Supplementary Table 3), is very close to that of free TAD, 0.46, verifying that the competitive pathway indeed involves free TAD.”

Additionally, as the competitive dissociation model involves free (unbound) TAD, using lower Mdm2 concentrations should allow for direct detection of the unbound state.

→ The difference in FRET efficiency between the Mdm2 bound TAD and free TAD is small (0.62 vs 0.46) and clear separation of these two states in binned trajectories is not possible for many transitions even at a lower Mdm2 concentration as shown in Fig. 2c and Supplementary Fig. 2b. Instead, the above maximum likelihood analysis can also determine the FRET efficiency of a potential intermediate state (with a lifetime of ~ 12 ms) along the competitive pathway. The FRET efficiency (the sum of the acceptor 1 and 2 fractions) was 0.43 and 0.48 at 200 nM and 2 μ M Mdm2, respectively, very close to 0.46 of free TAD. This clearly verifies that free TAD is involved in the competitive pathway. This result was added to the main text (page 5) and *Calculation of the relative flux involving the ternary complex* section in Methods (page 18) and Supplementary Table 3.

(page 5)

“... This pathway involves free, disordered TAD, but it was not observed in binned trajectories due to the high Mdm2 concentration, as mentioned above. A maximum likelihood analysis of photon trajectories near the transitions determined that the FRET efficiency of a short segment (~ 12 ms) of the TAD-Mdm2 state right before the transition to TAD-Taz2 is 0.48, very close to that of free TAD in the TAD-Mdm2 binding experiment above (Fig. 2d). This result verifies that the competitive pathway involves free TAD. A similar result was obtained from the experiment with $[Mdm2] = 200$ nM (see Calculation of the relative flux involving the ternary complex in Methods and Supplementary Table 3).”

(page 18)

“... This analysis also determines the acceptor fractions of the intermediate (~ 12 ms lifetime) between the TAD-Mdm2 and ternary complexes. The FRET efficiency (sum of acceptor 1 and acceptor 2 fractions, $\epsilon_1^{competitive} + \epsilon_2^{competitive}$ in Supplementary Table 3) of the intermediate along the competitive pathway, 0.48 (Supplementary Table 3), is very close to that of free TAD, 0.46, verifying that the competitive pathway indeed involves free TAD.”

8. The manuscript would benefit of more examples of curves and availability of code used for data analysis.

→ We are not entirely sure what the reviewer meant by more examples of curves. If the reviewer requested additional example trajectories, we provided more in Supplementary Figs. S2 and S3. In addition, we prepared raw data with codes for the data analysis and plots with library files, which will be shared if the paper is accepted. The analysis source codes can be found in the Github repository (https://github.com/hoisunglab/FRET_TransitionPath). Please see the data sharing statement.

Minor:

1. In Supplementary Figure 1, color code sequences that correspond to AD1 and AD2. It is unclear for the “full-length” TAD, if all the sequence not found in the AD2 construct corresponds to AD1. I would guess not as in Figure 1a, there is a long linker depicted in between each AD. Plus, the title of the Figure refers to NCBD, which is the protein that was studied in another manuscript.

→ The AD1 and AD2 regions are not clearly defined. The parts that fold and interact with different binding partners differ. And yes, there are natural flexible linker between the folded AD1 and AD2 regions as shown in the NMR structure of TAD-Taz2 (Fig. 1). We indicated the residues that form α helices in the AD1 and AD2 regions when TAD binds to Taz2 and Mdm2 in Supplementary Fig. 1. We thank the reviewer for pointing out the NCBD typo.

2. In Figure 1a, having AD1 and AD2 color coded the same as Mdm2 and Taz2, respectively is confusing. Would recommend to color code them as blue to be consistent with the rest of the figure. Polyethylene Glycol is misspelled on line 73.

→ Colors were changed in Fig. 1a and the typo was fixed.